# Ultrathin crystalline-silicon-based strain gauges with deep learning algorithms for silent speech interfaces

Taemin Kim [1,8], Yejee Shin[2,8], Kyowon Kang[1,8], Kiho Kim[1,8], Gwanho Kim[1,8], Yunsu Byeon[2,8], Hwayeon Kim[3], Yuyan Gao[4], Jeong Ryong Lee[2], Geonhui Son[2], Taeseong Kim[2], Yohan Jun[2,5,6], Jihyun Kim[3], Jinyoung Lee[3], Seyun Um[3], Yoohwan Kwon[3], Byung Gwan Son[3], Myeongki Cho[1], Mingyu Sang[1], Jongwoon Shin[1], Kyubeen Kim[1], Jungmin Suh[1], Heekyeong Choi [1], Seokjun Hong[1], Huanyu Cheng [4], Hong-Goo Kang[3] ✉, Dosik Hwang[2,7] ✉ & Ki Jun Yu [1,7] ✉

A wearable silent speech interface (SSI) is a promising platform that enables verbal communication without vocalization. The most widely studied methodology for SSI focuses on surface electromyography (sEMG). However, sEMG suffers from low scalability because of signal quality-related issues, including signal-to-noise ratio and interelectrode interference. Hence, here, we present a novel SSI by utilizing crystalline-silicon-based strain sensors combined with a 3D convolutional deep learning algorithm. Two perpendicularly placed strain gauges with minimized cell dimension (<0.1 mm²) could effectively capture the biaxial strain information with high reliability. We attached four strain sensors near the subject's mouths and collected strain data of unprecedently large wordsets (100 words), which our SSI can classify at a high accuracy rate (87.53%). Several analysis methods were demonstrated to verify the system's reliability, as well as the performance comparison with another SSI using sEMG electrodes with the same dimension, which exhibited a relatively low accuracy rate (42.60%).

The lack of clinical treatment for speech impediments caused by aphasia or dysarthria has been promoting various studies toward improving nonacoustic communication efficiency[1–5]. Silent speech recognition is one of the most promising approaches for addressing the above problems, in which facial movements are tracked by visual monitoring[6–10] or nonvisual capturing of various biosignals[11–14]. Visual monitoring, a well-known vision recognition, is the most direct method to map speech-related movements and has the highest spatial resolution[15,16]. However, there are many situations in which the daily use of vision recognition is limited since the continuous shooting of the face in a static environment is indispensable. Changes in the shooting direction due to body motion and changes in the light intensity according to the surrounding environment can lead to a significant drop in recognition accuracy. Furthermore, it is highly possible that unnecessary information such as background may occupy more pixels than speech-related information, resulting in an inefficient data processing. On the other hand, human–machine interfaces that exploit wearable electronics[1–5,12–14,17–23] for biosignal recording are used in a relatively dynamic environment. Electrophysiological signals, such as electroencephalography (EEG)[11,24–26], electrocorticography (ECoG)[27–29], and surface electromyography (sEMG)[12,30,31], have been extensively studied for SSI. Neural signals,

including EEG and ECoG, contain an enormous amount of information regarding brain activity in specific local regions that are activated during speech. However, EEG suffers from signal attenuation due to the skull and scalp[32], thereby impeding the differentiation of a large number of words driven by complex electrical activities[33]. By contrast, ECoG exhibits a much higher signal-to-noise ratio (SNR) compared to that of EEG, but it has limitation in clinical use because it is an invasive approach involving craniotomy. The sEMG, which measures electrical activities from facial muscles, can be extracted noninvasively and has relatively less complexity. Nonetheless, the low spatial resolution regarding SNR[34] and interelectrode correlation[35,36] hinder its application in the classification of a larger number of words. Furthermore, external issues, including signal degradation mainly due to body wastes, such as sweat and sebum alongside skin irritation, preclude long-term monitoring in real life[37].

Facial strain mapping using epidermal sensors provides another prospective platform to achieve a silent speech interface (SSI) with many advantages over all other existing systems. Various studies have explored the robustness of strain gauges in diverse facial movement detection applications[13,38,39], such as facial expression recognition and silent speech recognition. However, the large deformation of facial skin generated during expression or speech mostly relies on stretchable organic material-based strain sensors fabricated in a bottom–up approach[13,38,40]. These devices can make conformal contact with the skin and withstand tensile stress in severe deformation environments but suffer from their intrinsic device-to-device variation and poor long-term stability. These properties are critical drawbacks regarding deep learning-assisted classification because their repeatability is directly related to system accuracy. By contrast, inorganic materials, such as metals and semiconductors, are representative materials for fabricating strain gauges with high reliability and fast strain relaxation time leading to a fast respnse time of the gauges in a dynamic strain environment. The resistance of a conventional metal-based strain sensor varies according to the geometrical changes under the applied strain, resulting in a relatively low gauge factor (~2). However, for a semiconductor-based strain gauge, the piezoresistive effect is a dominant factor in the resistance change[41–43]. Under applied strain, the shift in bandgap induces carrier redistribution, thereby changing the mobility and effective mass of semiconductor materials. Because the resistance change caused by the piezoresistive effect is a few orders magnitude higher than that caused by the geometrical effect, the semiconductor-based strain gauge has an incomparable gauge factor (~100) to the metal-based strain gauge. Some of the latest silent communication systems based on various strain gauges are compared and summarized in Supplementary Table 1.

In this study, we propose a novel SSI using a strain sensor based on single-crystalline silicon with a 3D convolution deep learning algorithm to overcome the shortcomings of the existing SSI. The silicon gauge factor can be calculated using the equation: $G = (\triangle R/R)/(\triangle L/L) = 1 + 2\nu + \pi E$, where $\nu$ and $\pi$ are the Poisson's ratio and piezoresistive coefficient, respectively. Boron doping with a concentration of $5 \times 10^{18}$ cm$^{-3}$ was adopted to minimize resistance change due to external temperatures[44] while maintaining its relatively high piezoresistive coefficient (~80% of its value)[45]. High Young's modulus (E) of Si contributes to the fast strain relaxation time as well as sensitivity according to the equation: $T = \eta/E$, where $T$ is the relaxation time and $\eta$ is the viscous behavior term. However, since single-crystalline silicon exhibits inherent rigidity with a high Young's modulus (~160 GPa), stretchability must be achieved by modifying its structure into a fractal serpentine design[17,19,46]. Our epidermal strain sensor was fabricated with a self-standing ultrathin (Overall device thickness: <8 μm) mesh and serpentine structure without requiring an additional elastomeric layer, thereby providing enhanced air and sweat permeability[47], and comfort when attached. Additionally, we devised a biaxial strain sensor that can measure

directions and magnitudes in two dimension by placing two extremely small-sized (<0.1 mm$^2$) strain gauges in the horizontal and vertical directions, respectively. Based on a heuristic area feature study, four biaxial strain sensors were attached to the part where the skin deforms the most during silent speech. Because direct electrical contact is not required for strain measurements, our devices can leverage double-sided encapsulation, which delivers more secure protection of the active device layer, minimizing signal degradation caused by the aforementioned external factors such as sweat and sebum. Strain data of 100 words randomly selected from Lip Reading in the Wild (LRW)[48], each with 100 repetitions from two participants, were collected and used for deep learning model training. Our model with a 3D-convolution algorithm produced 87.53% recognition accuracy, which is an unprecedented high performance for this number of words compared with the existing SSIs using a strain gauge. Analysis of data measured over multiple days from two subjects suggested that our system captured the signal of each user's word characteristics despite the sensor location dependency and user dependency. We believe that this result is comparable with the state-of-the-art result of the SSI using the sEMG dry electrode, whose dimension is approximately two orders of magnitude exceeding our strain gauge[12]. We also fabricated an sEMG sensor with identical dimensions as our strain gauge, which exhibited much lower recognition accuracy, 42.60%, compared to that of our system. This comparison verifies the advantage of our system's high scalability, facilitating extended word classification.

## Results and discussion
### Overview of SSI with strain sensors

Figure 1a shows the stacked structure of our stretchable sensor embedding two silicon nanomembrane (SiNM)-based strain gauges (thickness ~300 nm) located perpendicular to each other in flexible polymer layers. The total thickness of the fabricated device was less than 8 μm, enabling the conformal attachment to the skin when a water-soluble tape was used as a carrier of temporary tattoo. During silent speech, muscle movements around the mouth induce skin deformation, which can be precisely monitored using perpendicularly placed strain gauges. Highly sensitive SiNM-based strain gauges and flexible polyimide film have relatively high Young's moduli of approximately 130 and 1 GPa, respectively, making them inappropriate candidates for stretchable devices. Therefore, the whole components of our sensor are patterned into mesh and serpentine structures to achieve stretchability and long-term stability for this application[17,19,46].

Each part of the face skin differs in stretching degree and direction when speaking silently, depending on the targeted words. Accordingly, determining proper sensor locations significantly contributed to SSI performance. An auxiliary vision recognition experiment that extracts the area features of the face was conducted for this purpose (Supplementary Fig. 1a). Among the randomly partitioned 24 compartments around the mouth, the sections with larger areal changes during silent speech were assumed to involve more strain gradients. Relevance-weighted class activation map (R-CAM) analysis revealed significant changes in the areas just below the lower lip (Sections 1, 4, 5, and 9 in Supplementary Fig. 1b)[49]. In addition, the ablation study revealed that significant differences were unobserved in the recognition accuracy between acquiring data from the half side and both sides of the face, as the facial skin moved almost symmetrically during silent speech (Supplementary Fig. 2a). In order to obtain the most dynamic strain siganls from the skin and the positions at which attachments of the devices are convenient, the four sites were determined as S1(A), S2(B), S3(C), and S4(D) (Fig. 1b), matching Sections 15, 16, 20, and 24, respectively, in Supplementary Fig. 2b.

Figure 1c shows that the four strain sensors, each incorporating two SiNM gauges, captured the resistance change in the time domain through eight independent channels when a word such as

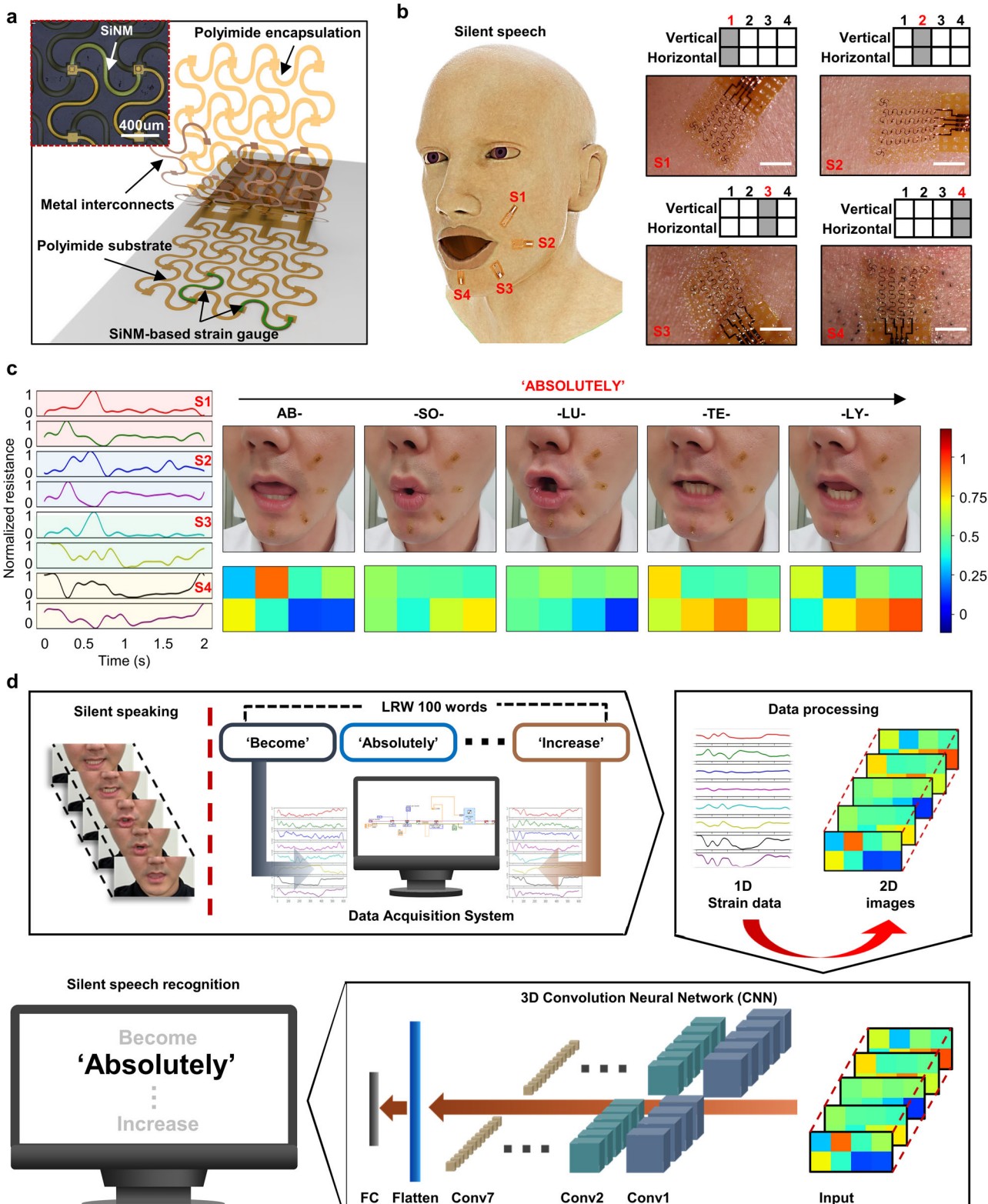

**Fig. 1 | Silent speech recognition system using strain sensors and deep learning. a** Expanded view of single-crystalline silicon nanomembrane (SiNM)-based stretchable strain sensor; the thickness of transfer-printed SiNM is 300 nm. Metal interconnect comprises a thin layer of Au(250 nm)/Cr(5 nm) deposited by thermal evaporation. Both substrate and encapsulation layers are made of spin-coated polyimide double-layer (thickness: 3.4 μm). Inset: optical microscopic image of a unit cell of strain gauge. **b** Three-dimensional modeling of a human face wearing four devices. Each device is integrated with two strain gauges, a vertical gauge, and a horizontal gauge. Inset: photographs of the deformation of each sensor during silent speech of vowel A; scale bars, 2 mm. **c** Waveform and heatmaps of the resistance changes from 8-channels strain gauges with respect to time during the pronunciation of the word, ABSOLUTELY. **d** Overall flowchart of silent speech recognition system, including strain DAQ, data preprocessing, feature extraction, and word classification. The 100 words used in this study are randomly selected from LRW-1000, which is generally considered a benchmark in word recognition.

ABSOLUTELY was silently pronounced. Different resistance changes were monitored at each channel according to the varying shape of the subject's mouth by mapping the normalized resistance changes at each channel over time into a $2 \times 4$ heatmap. Concatenating these matrices according to the time sequences, the targeted words can be digitized into a three-dimensional (3D) matrix containing each designated position and time information.

Figure 1d shows the overall flow of the hardware and software processes of our SSI. In this study, when an enunciator silently uttered a random word out of the 100 words, strain information from the eight channels was recorded by a data acquisition (DAQ) system (Supplementary Figs. 3 and 4). Considering the positional correlation between the biaxial gauges, the 1D signal data were processed as sequential 2D images as input data. We adopted a 3D convolutional neural network (CNN) to encode spatiotemporal features from an SiNM strain gauge signal. We trained our network with five-fold cross-validation and analyzed how it makes decisions based on explainable artificial intelligence.

### Hardware characterization of the biaxial strain sensor

The facial skin expands and contracts in all directions based on a specific point when a person speaks. Therefore, the degree of skin extension and information on the direction are necessary for accurate tracking of facial skin movement. Here, we designed a biaxial strain sensor that independently quantified the strain in two mutually orthogonal directions by integrating a pair of strain gauges positioned in the horizontal and vertical directions, respectively (Fig. 2a).

To characterize the electrical properties of the SiNM-based strain gauge, uniaxial tensile stress was applied on the x- and y-axis up to 30%, considering the elastic limit of the facial skin during silent speech[50]. Finite element analysis (FEA) of the strain distribution demonstrates that a horizontal gauge experiences much higher strain with 30% x-axis stretching compared to vertical gauge, and vice versa with 30% y-axis stretching (Fig. 2b, c). This result corresponds with the actual uniaxial stretching test. Supplementary Note 1 and Supplementary Table 2 detail the Piezoresistive Multiphysics model used in FEA. Figure 2d, e shows the relative resistance change of the horizontal and vertical gauges, respectively, showing a stepwise increase regarding the increment in applied strain. When a collateral force was applied to the strain gauge, it induced a dominant resistance change, whereas the orthogonal force induced a relatively small resistance change. Along with high sensitivity, reliable DAQ is important for SSI applications. To confirm our sensor repeatability, a cyclic stretching test was also conducted by attaching the device to an elastomer with a modulus comparable with that of human skin. Even after 50,000 cycles of 30% stretching, our strain sensor showed negligible change in its resistance, confirming its high reliability (Fig. 2f).

A metal-based strain gauge with an identical structure to an SiNM-based strain gauge was also fabricated to check the feasibility of this application. Figure 2g shows the comparison of relative resistance changes between SiNM-based and metal-based gauges while stretching up to 30%. The result showed that the SiNM-based gauge was approximately 42.7, 28.9, and 20.8 times more sensitive for 10%, 20%, and 30% stretching, respectively, than those of the metal gauge. Figure 2h shows the captured relative resistance change of two gauges for eight channels while silently pronouncing the same word WITHOUT Through its high gauge factor, the SiNM-based gauge exhibited a remarkable waveform, whereas the metal-based gauge showed almost indistinguishable changes. For a normalized waveform, which is an input form for feature extraction, the SiNM-based gauge exhibited a distinct resistance change between 0.5 and 1.5 s, whereas no conspicuous change was monitored because of the similar level of noise for metal-based gauge (Fig. 2i).

### Three-dimensional CNN for SiNM strain gauge signal analysis

Our goal was to classify the 100 words from the SiNM strain gauge signals with a time length of 2 s measured at 300 frames per second. To utilize both spatial and temporal information, we used a 3D CNN model for the classification task. Figure 3a illustrates the detailed architecture of the model. Our model comprised seven 3D convolution layers and three fully connected (FC) layers. We used the kernel size of (3,3,3), padding (1,1,1), and stride (1,1,1) except the Conv3 layer where we used the kernel size of (3,1,3), padding (1,0,1), and strides (2,1,2) for downsampling. For each layer, we used instance normalization and ReLU activation. The pooling layer was not used to preserve localized spatial information. We flattened the output features of the last convolution layer (Conv7), then it was connected to several FC layers for classification. We used cross-entropy loss and the Adam optimizer[51] to train our 3D CNN. More details are provided in Supplementary Table 3.

### Results of silent speech recognition

We performed a word classification task with our SSI system to 100 datasets per 100 words recorded by two subjects (See Supplementary Table 4 for details). To provide an insight on the generalized performance of our proposed system to an independent dataset, we performed five-fold cross-validation tests with randomly mixed datasets. Figure 3b shows the results. The accuracy of the five-fold cross-validation test ranged from 80.1% to 91.55%, and the average was 87.53%. We also evaluated word classification accuracy by varying the number of trained data, and compared the results with a conventional support vector machine (SVM)-based classification model (Fig. 3c)[52]. Not surprisingly, the accuracy of the FOLD 5 validation set test improved as the trained data increased, from 23.70% with 10 cases to 87.50% with 80 cases. Our model showed at least 15% higher accuracy than the SVM model when the number of trained dataset is larger than or equal to 20. The performance comparison with other models which are commonly used to handle sequential data such as speech/audio, text, and video is provided in Supplementary Table 5. We also investigated the performance variation depending on the number of sensors used. As shown in Fig. 3d, word accuracy improved from 49.87% to 87.53% as the number of channels increased from 2 to 8. We obtained these results by averaging all the feasible combinations of horizontal and vertical channels.

To evaluate our SSI performance, we compared it with the following classifier models: correlation and SVM. Figure 3e shows the confusion matrix of the recognition results for 20 datasets (Fold 3) using these classifier models. The correlation model used one target dataset as a reference, and the results were calculated using the cosine similarity of each word between the reference and other datasets. This experiment predicted words with the highest similarity scores. We repeated this operation by changing the reference dataset. Figure 3e shows the results obtained by averaging all the cases. Our proposed method's accuracy reached 91.55% for the FOLD 3 validation set, significantly exceeding those of the correlation and SVM (average accuracy: 10.26% and 76.30%, respectively). Supplementary Tables 6 and 7 present the accuracy per word of our SSI. Furthermore, we evaluated the performance variation of our model to unseen data, of which accuracy may drop due to the mismatch of sensor location and subject dependency. Although the unseen datasets taken from the completely different domain from the test datasets were used, the classification accuracy could gradually be improved if we adapted the model using a transfer learning, which increased the accuracy sharply even up to 88%. This demonstrated that our sensors extracted meaningful values even if the attached points could be slightly misplaced. Supplementary Tables 8 and 9 detail the accuracy of the results.

### Visualization

To visualize the high-dimensional features learned from deep learning models, we utilized t-distributed stochastic neighbor embedding (t-

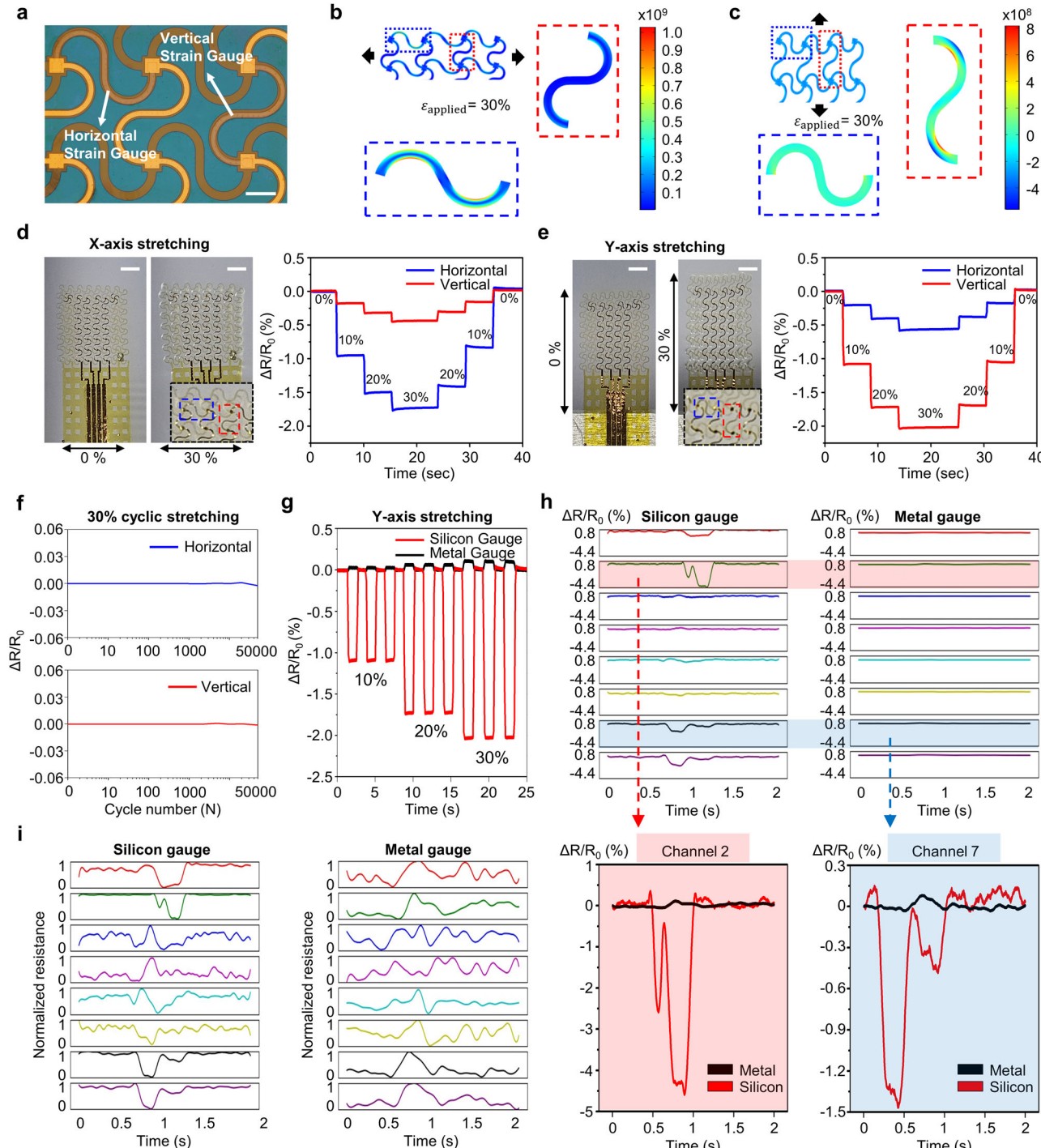

**Fig. 2 | Characterizations of SiNM-based biaxial strain gauge. a** Magnified optical image of biaxial strain gauges comprising a horizontal gauge and a vertical gauge; scale bar, 300 μm. **b**, **c** Finite element analysis of applied local strain to the biaxial strain gauge on an elastomer substrate with 30% stretching along x (**b**) and y (**c**) directions. **d**, **e** Photographs of the biaxial strain gauges before and after 30% stretching test (left) and resistance change of both horizontal and vertical gauges during in vitro test at 10%, 20%, and 30% stretching (right) in x (**d**) and y (**e**) directions, showing independent sensing properties of the biaxial strain gauges where the stretching in parallel and perpendicular directions to the gauge is prone to apply dominant and minor strain, respectively; scale bars, 1 mm. Inset: enlarged photos of a biaxial strain gauge under an applied strain. **f** Relative change in electrical resistance during 50000 cycles of 30% stretching along y direction under 10 mm/s. Each cycle has a start delay and end delay of 1 s. **g** Comparison of sensitivity to strain between SiNM- and metal-based strain gauge through 10%, 20%, and 30% cyclic stretching test. **h** Waveforms of corresponding in vivo test of both SiNM- and metal-based strain gauges during silent speech of the word, WITHOUT (**h**, top), and magnified plots of channels 2 (red highlight) and 7 (blue highlight) (**h**, bottom). **i** Normalized waveforms of **h** in the training phase.

SNE)[53], which is commonly used to map high-dimensional features into two- or three-dimensional planes. We visualized high-dimensional feature outputs of the 3D convolutional deep learning model in two dimensions (Fig. 4a and Supplementary Fig. 5). The t-SNE results for

the 100 classes of the test dataset showed that each class was well grouped together. We selected 10 specific classes out of the 100 words and visualized them for further analysis (Fig. 4b). Observably, the words with similar pronunciations were mapped to be close to each

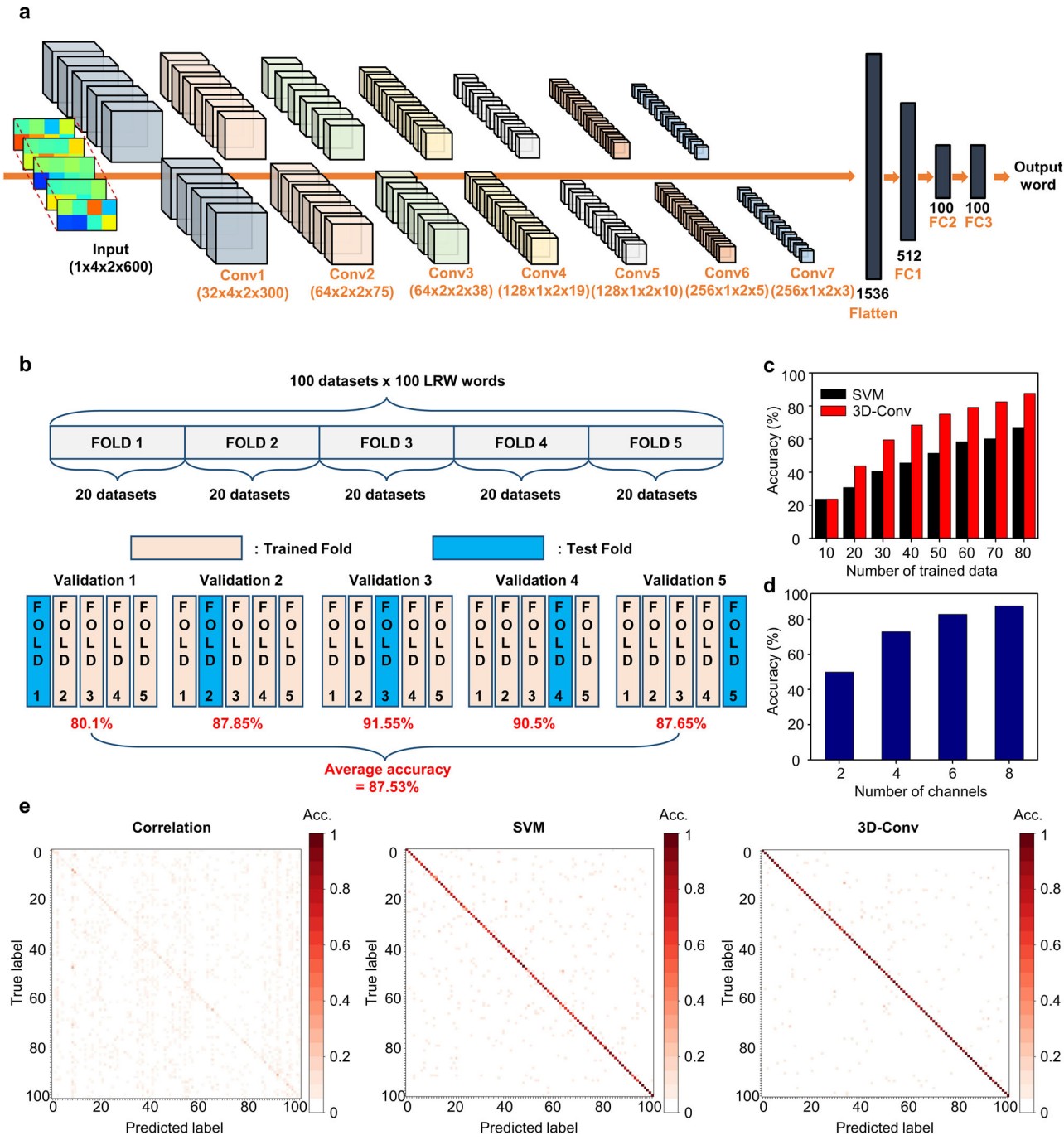

**Fig. 3 | Method and validation results of silent word recognition. a** Pipeline of our deep learning model architecture comprising mainly 3D convolutional layers. **b** Procedure of evaluating the proposed silent speech recognition system. 100 datasets, comprising 100 words each, are randomly divided into five folds and cross-validated; 58 and 42 datasets out of 100 datasets are acquired from different subjects: A and B, respectively. **c** Comparison of the recognition performance of two different classifier models, SVM and our deep learning model, as the number of trained data increases. Each accuracy rate is the average value of five independent validations where FOLD 5 in **b** is fixed as a test dataset, and n datasets randomly selected from the other four folds are trained in our deep learning model. **d** Word recognition rates in the number of sensor channels. Each accuracy of n channels out of eight channels is the arithmetic mean of the accuracies from all the n-combinations of the eight channels ($_8C_n$) set. **e** Confusion matrices of word prediction results from three different classifier models, including correlation (left), SVM (middle), and 3D convolution (right), with the average accuracy rates of 10.26%, 76.30%, and 87.53%, respectively.

other (INCREASE vs DEGREES and FAMILY vs FAMILIES). Supplementary Tables 6 and 7 summarize quantitative results obtained by these confusing words. The raw signal waveform of these similar words resembled each other (Fig. 4c). Therefore, it is inevitably difficult for the word-based classification model to distinguish between these similar pronounced words. Notably, our model can provide correct classification by detecting changes in the muscles around the mouth.

We analyzed the characteristic of our deep learning-based classification model through R-CAM[49], a method for visualizing how much each region is affected by a classification task. Figure 4d illustrates the R-CAM results to the words, ABSOLUTELY and AFTERNOON. For both words, our model focused on the part in which the S2 sensor signal (third- and fourth-row signals) showed dominant characteristic movements. Regarding the word ABSOLUTELY, our model focused on

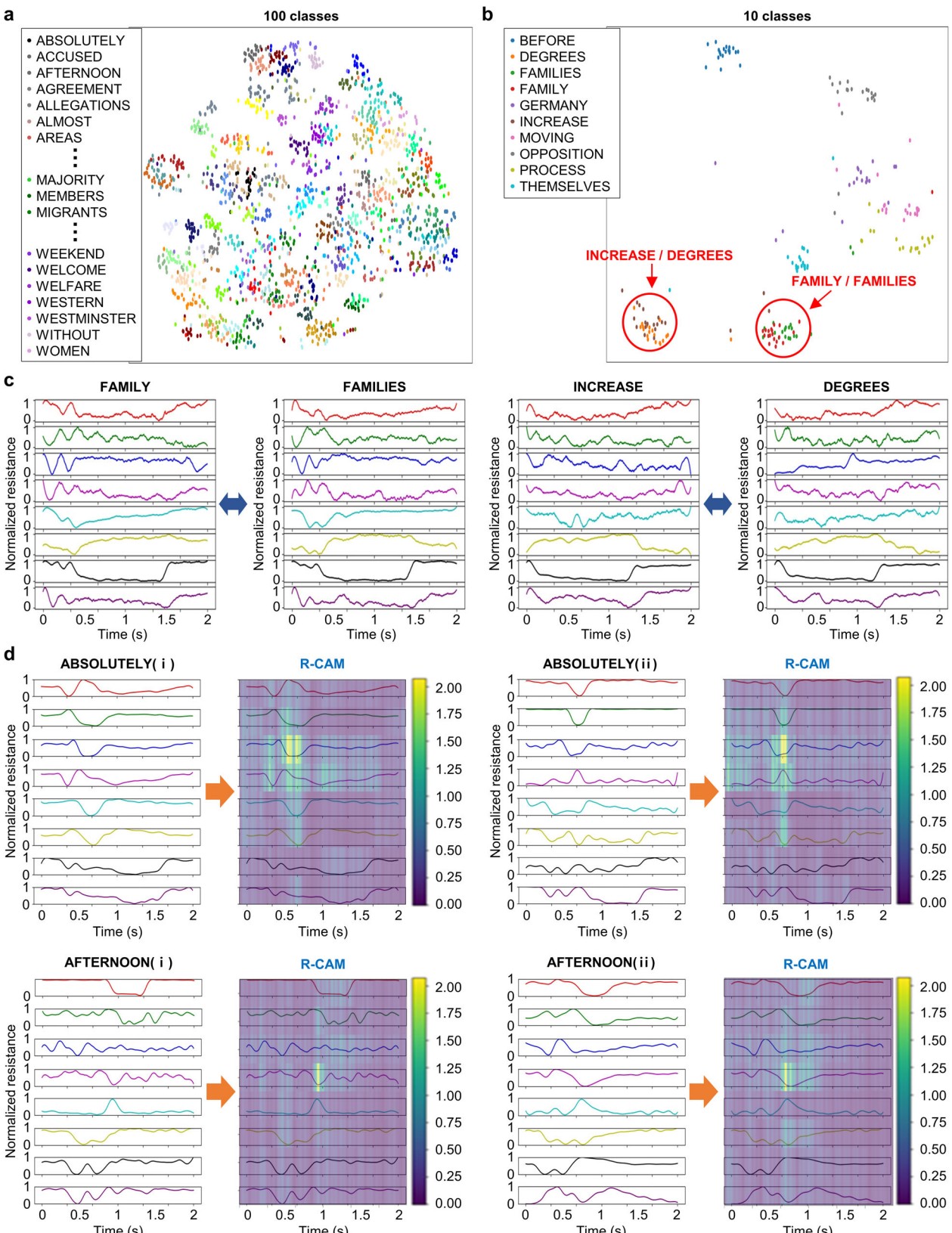

**Fig. 4 | Analysis and verification of results. a** T-distributed stochastic neighbor embedding (t-SNE) of 100 words, visualizing the result of validation 1 in Fig. 3b. All 100 words are allocated in points of different colors. The denser the cluster of same-colored points is plotted, the more the model classifies them as similar data. **b** t-SNE of the most confused 10 words out of 100 words. **c** Two pairs of normalized waveforms of words exhibiting similar facial movement during pronunciation (FAMILY/FAMILIES and INCREASE/DEGREES). **d** Relevance-CAM (R-CAM) that explains our model by highlighting which region of the entire waveform is dominant to classify each word, demonstrating that the model focuses on characteristic signal parts where the variance of the resistance is large.

the downward and upward convexities of sensor S2 at the time of 0.6 s. Concerning AFTERNOON, similarly, our model focused on the downward convex point in both cases, which is at around 1 s for AFTERNOON(i) and at around 0.7 s for AFTERNOON(ii). The results demonstrated that our model was not overfitted to signal data but focused on characteristic signal parts where the resistance variance was large.

### Comparison of word recognition performance with sEMG

As shown in Fig. 5a–c, three types of epidermal sEMG electrodes with various dimensions were also fabricated to determine the dependence of electrode size to acquired signal quality. The surface area of the small-sized electrode was nearly identical to the unit cell of our strain gauge so that we could fairly compare the scalability of the two systems, whereas those in medium- and large-sized electrodes were comparable with the conventional epidermal sEMG electrodes for other SSIs[12,31]. A pair of 2-channels sEMG electrodes and one commercial EMG reference electrode were attached to the buccinators and near the posterior mastoid, respectively, and the sEMG signal was obtained at a sampling frequency of 1 kHz when the subject's jaw was clenched. The raw sEMG signal was preprocessed with a commercial EMG module comprising three filters and an amplifier before being transmitted to a DAQ module (Supplementary Fig. 6). The calculated SNR (1.517, 5.964, and 8.378 for small-, medium-, and large-sized electrodes, respectively) increased as the electrode dimension increased because of the lowered surface impedance (see Fig. 5d–f), revealing the limitation in improving the spatial resolution of sEMG data.

To compare the word classification accuracy of sEMG-based model with that of our stran gauges-based system, four pairs of small-sized sEMG electrodes were attached to the facial muscles, which are generally selected for SSI, including buccinators, levator anguli oris, depressor anguli oris, and the anterior belly of digastric (Supplementary Fig. 7)[12]. As in the case of DAQ using our strain gauge, 100 datasets of sEMG signals were obtained from the two subjects when silently speaking 100 words, followed by hardware and software signal processing (Supplementary Fig. 6). The preprocessed datasets were randomly partitioned into five folds, and each fold feature was extracted and cross-validated using the same method (see the flowchart in Fig. 3b). Figure 5g shows the confusion matrix of classification results, where the average recognition accuracy was 42.60%. The state-of-the-art performance[12] used the system electrode whose size is two orders of magnitude larger than that of this work. To make a fair comparison, we downscaled the sEMG electrodes to be identical size of our strain gauges. With the sEMG waveforms from 100 words data as inputs, the feature embeddings output by the deep learning model are shown in Fig. 5h. The 2D t-SNE mapping showed that the points with the same color were scattered rather than clustered at a specific location, indicating the difficulty in learning the representation of the scattered raw data information. Supplementary Fig. 8 shows the magnified t-SNE plot with labeling of 100 words. This result, probably due to the diminished SNR, symbolized the impeding factor of sEMG for extended word recognition because more data with high spatial resolution induce a higher classification accuracy of extended wordsets.

## Conclusion

In summary, single-crystalline silicon-based strain gauges with a mesh and serpentine structure could be a promising candidate for silent speech communication with high scalability. Controlled doped single-crystalline silicon, having the advantages of high gauge factor and stability as an inorganic material, establishes a more accurate system for SSI through deep learning model training with high reliability and repeatability. The FEA simulation and automatic stretching test results demonstrated that four sets of two adjacent gauges positioned perpendicular to each other are suitable for measuring the two-dimensional movement of the skin. Additionally, we demonstrated that the silicon-based strain gauge provides superior sensitivity compared to the metal-based one with the same structure under a strain of 30%. Coupled with a novel 3D convolution deep learning model, we achieved a word recognition accuracy of 87.53% to 100 words with eight strain gauges, whereas eight EMG electrodes with the same dimensions as ours only yielded an accuracy of approximately 42.60%. These results suggest a new platform by scaling the number of channels of the sensor system for SSI with a high spatiotemporal resolution, thereby providing a phoneme unit recognition capability that was previously impossible with any other systems.

## Methods

### Materials

SOITEC supplied SOI wafers (300 nm Si/1000 nm $SiO_2$), and KAYAKU Advanced Materials supplied 495 PMMA A8. Polyamic acid solution (12.8 wt%; 80% NMP/20% aromatic hydrocarbon) was purchased from Sigma-Aldrich. Two positive photoresists used for the photolithography process, MICROPOSIT S1805 and AZ 5214E, were from DOW and MicroChemicals, respectively. A photoresist developer and AZ 300mif from MicroChemicals were used for both photoresists. All the materials for cleaning (HF solution, buffered oxide etchant 6:1, sulfuric acid, and hydrogen peroxide) were purchased from REAGENTS DUKSAN. The PDMS base and curing agent, Sylgard 184, were purchased from DOW. Cr etchant (CT-1200S) and Au etchant (AT-409LB) were purchased from JEONYOUNG. Cu etchant (CE-100) was purchased from the Transene Company.

### SiNM transfer process

First, the SOI wafer was deep cleaned using piranha solution ($H_2SO_4$:$H_2O_2$ = 3:1) at 100 °C for 15 min and buffered oxide etchant for 5 s, followed by boron doping (high energy implantation, Axcelis) at a dose of 5e14 $cm^{-2}$, followed by rapid thermal annealing (RTA200H-SP1, NYMTECH) at 1050 °C for 90 s. The above cleaning process was repeated once more after the doping process. Second, microholes with 3 μm diameter and 50 μm pitch were defined throughout the device layer of the SOI wafer via UV–lithography (MDA-400S, Midas System) and reactive ion etching (Q190620-M01, Young Hi-Tech). MICROPOSIT S1805 was used as a positive photoresist for better adhesion with an elastomer stamp due to its high surface uniformity. Hole-patterned SOI wafers were then immersed in the HF solution for 25 min to dissolve the BOX layer and to release the device layer from the handle substrate. After rinsing with DI water, the released SiNM was transferred to the elastomer stamp (PDMS base: curing agent = 4:1) with moderate pressure, and the stamp was then pressed onto a PI layer soft-baked at 110 °C for 1 min. After baking at 150 °C for 3 min, the transfer printing process was completed by removing the stamp and photoresist.

### Biaxial strain sensor fabrication

The fabrication process started with preparing the two substrates on a silicon thermal oxide wafer cleaned using a piranha solution. A 500 nm-thick PMMA was spin coated and baked at 180 °C for 3 min as a sacrificial substrate to release completed devices after the whole fabrication process. Subsequently, a thin film of a PI double-layer (~3.4 μm) was formed as a supporting substrate by spin coating of liquid polyamic acid solution on the PMMA sacrificial layer. The PI substrate was then fully baked in a vacuum oven at 210 °C for 2 h after the transfer printing process of the SiNM layer. Two gauges perpendicular to each other were defined by photolithography and RIE. For better electrical contact with metallization, SiNM-based gauges were cleaned with buffered oxide etchant to remove the native oxide layer. Thermal evaporators (KVE-T2000, Korea Vacuum Tech) were used to deposit the metal layer of Au(250 nm)/Cr(5 nm) followed by UV–lithography with AZ-5214E positive photoresist to avoid overetching and then wet

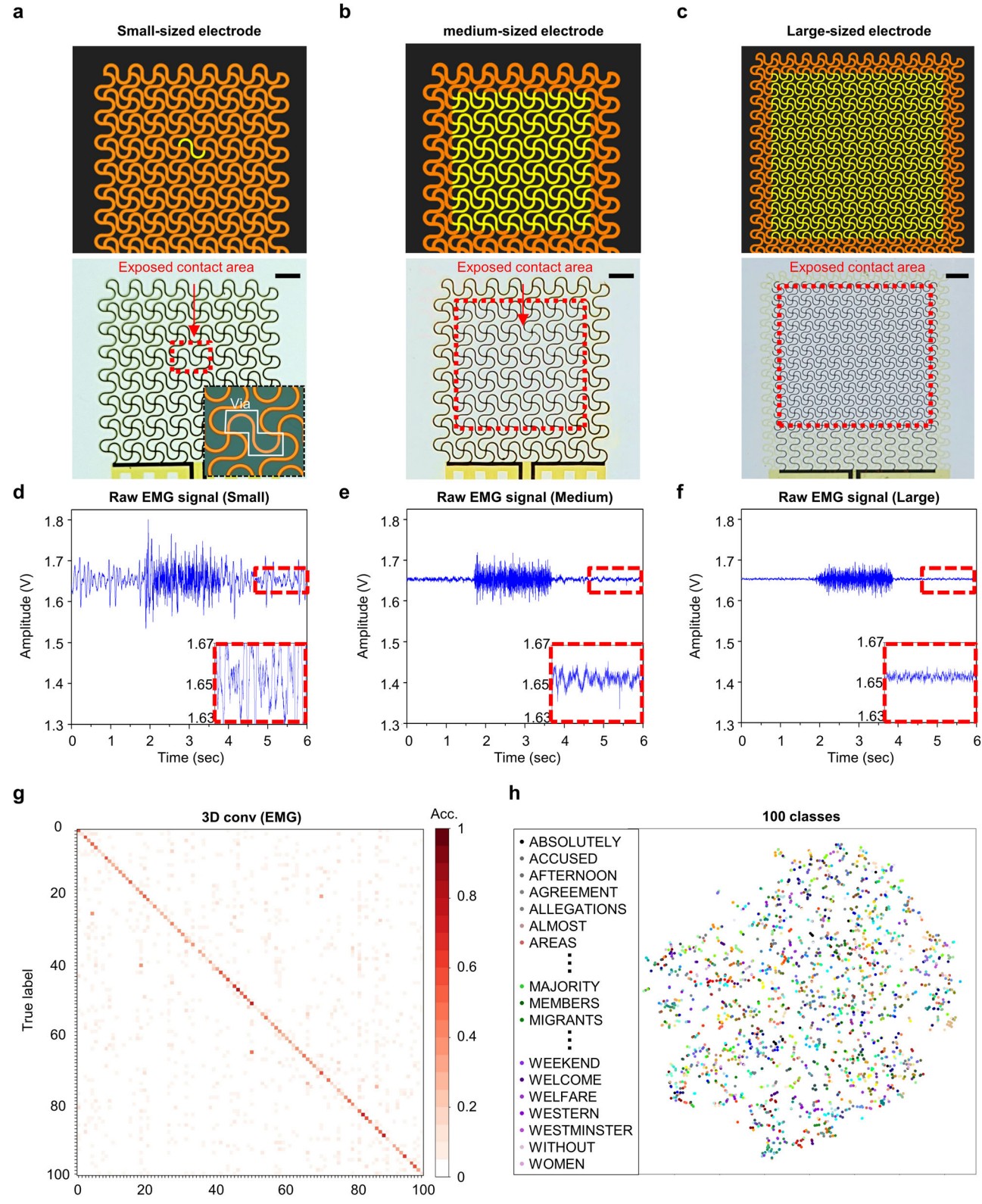

**Fig. 5 | Control experiment of silent word recognition using EMG. a–c** Schematics (top) and photographs (bottom) of three epidermal sEMG electrodes with different electrode dimensions. The exposed contact areas of small-sized (**a**), medium-sized (**b**), and large-sized (**c**) electrodes are ~0.1, ~5.5, and ~22.3 mm², respectively. Other than the exposed contact area is encapsulated with a polyimide layer. Inset in (**a**, bottom): magnified optical image of small-sized electrode; scale bars, 1 mm (**a**, **b**) and 1.5 mm (**c**). **d–f** Raw EMG signals of three sEMG electrodes while the subjects clench their jaw tightly with electrodes attached to the buccinators. Insets: magnified views of noise part. **g** Confusion matrix of the result of silent 100 words recognition using our small-sized EMG sensor (four channels with eight electrodes on buccinators, levator anguli oris, depressor anguli oris, and anterior belly of digastric) with the recognition rate of 42.60%. A total of 100 datasets are acquired, 46 of which are from Subject A and 54 are from Subject B. **h**, t-SNE of the total 100-word classification.

etching. After spin coating and curing the additional PI double-layer (~3.4 μm) for the encapsulation layer, 150 nm of the Cu mask layer was deposited and patterned to define the mesh and serpentine design. The whole structure of the device was then dry etched according to the etch mask, and the Cu mask was then wet etched. Afterward, the PMMA sacrificial layer was dissolved by immersing it in an acetone bath, and the released device was transferred to the water-soluble tape. Supplementary Fig. 9 illustrates the schematics of the fabrication process. For the metal-based strain sensor, the whole sequence was the same, except that the SiNM transfer and metal wet etching processes were substituted for deposition of Au(50 nm)/Cr(5 nm) and liftoff, respectively. Step-by-step fabrication process is detailed in Supplementary Note 2.

### sEMG electrode fabrication

As with the strain sensor above, the fabrication process of sEMG electrodes started with a coating of the PMMA sacrificial layer and a subsequent thin film of the PI layer (1.7 μm) on a cleaned silicon thermal oxide wafer. An electrode layer of Au(160 nm)/Cr(5 nm) was deposited by thermal evaporation and then patterned via UV–lithography and wet etching. An additional layer of PI (1.7 μm) for passivation was spin coated followed by thermal deposition of a Cu(100 nm) mask layer, as mentioned before. The unmasked area was dry etched using RIE, providing a mesh and serpentine design. Through this process, the designated active area was simultaneously exposed to direct contact with the skin (Fig. 5a–c). Finally, the device was immersed in an acetone bath to remove the underlying PMMA layer, resulting in device detachment from the handle substrate and subsequent transfer to a water-soluble tape. Supplementary Fig. 10 illustrates the schematics of the fabrication process, and step-by-step fabrication process is detailed in Supplementary Note 3.

### Experimental process of strain DAQ

Before attaching the strain sensor, the targeted skin was cleaned with ethanol and water. A skin-safe pressure-sensitive adhesive (Derma-tac from Smooth-On) was applied to the backside of the sensors on water-soluble tapes. And then, the sensors were attached to the position which was selected through the preliminary study presented in Supplementary Figs. 1 and 2 with moderate pressure, and DI water was then gently sprayed using a dispenser for 1 min to dissolve the PVA film. Residues of water-soluble tape were carefully peeled up using a tweezer. The whole device attachment process is demonstrated in Supplementary Movie 3. The strain sensors were connected to the breadboard comprising voltage divider circuit components by a pre-soldered ACF cable and jumper wires. The voltage divider provided $V_{in}$ from a 3 V common supply voltage generated by a voltage output DAQ module (PXIe-6738 from NI). $V_{in}$ was then measured with a voltage input DAQ module (PXIe-6365 from NI) with a 300 Hz sampling frequency. The flowchart and experimental setting image of the strain DAQ system with the voltage divider circuit are shown in Supplementary Figs. 3 and 4. The demonstrations of word recognition process are provided in Supplementary Movie 1 and 2.

### Experimental process of sEMG DAQ

For an unbiased comparison of the two SSIs, the DAQ of sEMG was performed following previous literature with state-of-the-art performance. A pair of sEMG electrodes on a water-soluble tape was transferred to a 3 M Tegaderm with 2 cm spacing, followed by the removal of the water-soluble tape in a temporary tattoo-like manner. By using Tegaderm, a transparent dressing adhesive film directly attached onto the backside of the sEMG sensor, the sEMG electrode areas located on the front side are fully opened to make direct contact to the skin while elsewhere is firmly attached to the skin. After cleaning the allocated locations with ethanol and water, four pairs of electrodes were attached to the adhesive of Tegaderm (Supplementary Fig. 10). The

reference electrode was attached near a posterior mastoid, which was electrically neutral from the sEMG measurement sites. All the electrodes were connected to commercial sEMG modules (PSL-iEMG2 from PhysioLab), incorporating three filters and an amplifier. The obtained sEMG signals were then carried to the voltage input DAQ module (PXIe-6365, NI) with a 1000 Hz sampling frequency. Supplementary Fig. 6 shows the details of the DAQ process.

### Software environment and the SSI process

The environment was based on Ubuntu 18.04. CUDA 11.2, anaconda3, and python 3.8 were installed. Adjacent location values were located sequentially to reflect the geometric characteristics in which they were correlated with each other. Signals with a video as input are $X \in \mathbb{R}^{1 \times H \times W \times T}$, where 1 is the number of videos, and T is the number of frames. H × W is the size of the frame where H is 2 with the paired horizontal and vertical axes, and W is 4 with the number of the location of strain gauge sensors. To extract the properties of resistance changes, we used min–max normalization for each signal and applied a Savitzky–Golay filter to reduce noise. The preprocessed signal data were fed into the 3D convolution-based model to consider spatio-temporal information. The detailed structure of each model is provided in Supplementary Tables 10–13.

### Ethical information for human subjects

According to Article 13, Paragraph (1) of the Enforcement Regulations of the Bioethics and Safety Act of the Ministry of Health and Welfare, Korea, the authors confirmed that there is no need to obtain Institutional Review Board approval as volunteers have conducted research using wearable sensors and simple contact measuring equipment without any physical modification nor invasive measurement on human. The authors affirm that human research participants provided informed consent for publication of the images in Fig. 1c and Supplementary Figs. 1a, 2, and 7.

## Data availability

All datasets generated during this study are available from the corresponding author upon reasonable request. The raw data of device characteristics are provided in the Source Data files. The silent speech data for 100 LRW words generated by strain and sEMG DAQ system are available at the following link. https://drive.google.com/file/d/1UssIGck1sy9wDtiSYt_YrEM59bx6nPIc/view?usp=sharing. Source data are provided with this paper.

## Code availability

All codes used in this study are available at the following database. https://github.com/MAILAB-Yonsei/Silent-Speech-Interfaces.

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

## Acknowledgements
This work was supported by Samsung Research Funding Center of Samsung Electronics under Project Number SRFC-IT1901-08.

## Author contributions
Taemin K., Y.S., Kyowon K., Kiho K., G.K., and Y.B. contributed equally to this work. Taemin K., Kyowon K., Kiho K., and G.K. designed the devices and performed the hardware experiments. Y.S. and Y.B. designed the deep learning model. Taemin K., Kyowon K., Kiho K., G.K., M.C., M.S., Jongwoon S., Kyubeen K., Jungmin S., Heekyeong C., S.H. fabricated the devices. Y.G. and Huanyu C. performed the FEA simulations. J.R.L. and G.S. performed the software experiments and analyzed the data. Tae-seong K. and Y.J. were participated in the discussion of the methods. H.K. and B.G.S. preprocessed the strain sensor signal. J.K., J.L, S.U., and Y.K. did a pilot study about silent word recognition. K.J.Y., D.H., H.-G.K. supervised and directed this work. Taemin K., Y.S., Kyowon K., Kiho K., G.K, Y.B., J.R.L, G.S., Taeseong K., Y.J., H.-G.K., D.H., K.J.Y. wrote the manuscript. All authors discussed and commented on the manuscript.

## Competing interests
The authors declare no competing interests.

## Additional information

[1]Functional Bio-integrated Electronics and Energy Management Lab, School of Electrical and Electronic Engineering, Yonsei University, 50 Yonsei-ro, Seodaemun-gu, Seoul 03722, Republic of Korea. [2]Medical Artificial Intelligence Lab, School of Electrical and Electronic Engineering, Yonsei University, 50, Yonsei-ro, Seodaemun-gu, Seoul 03722, Republic of Korea. [3]Digital Signal Processing & Artificial Intelligence Lab, School of Electrical and Electronic Engineering, Yonsei University, 50, Yonsei-ro, Seodaemun-gu, Seoul 03722, Republic of Korea. [4]Department of Engineering Science and Mechanics, The Pennsylvania State University, University Park, PA 16802, USA. [5]Athinoula A. Martinos Center for Biomedical Imaging, Massachusetts General Hospital, Charlestown, MA, USA. [6]Department of Radiology, Harvard Medical School, Boston, MA, USA. [7]Department of Electrical and Electronic Engineering, YU-Korea Institute of Science and Technology (KIST) Institute, Yonsei University, 50, Yonsei-ro, Seodaemun-gu, Seoul 03722, Korea. [8]These authors contributed equally: Taemin Kim, Yejee Shin, Kyowon Kang, Kiho Kim, Gwanho Kim, Yunsu Byeon. ✉e-mail: hgkang@yonsei.ac.kr; dosik.hwang@yonsei.ac.kr; kijunnyu@yonsei.ac.kr

