## [Peer Review File · Nature Communications]

REVIEWER COMMENTS

Reviewer #1 (Remarks to the Author):

The manuscript is of good quality overall and shows interesting results, with a few shortcomings. It can be published following a revision to clarify the following aspects and add the missing information.

The comparison between semiconductor and with metal-based strain gauges (lines 238-240) is not sufficiently informative of the state-of-the-art in relation to strain gauges-based systems. It would be useful to compare with other wearable state-of-the-art strain gauges which have already being used for similar purposes, such as graphene-based strain gauges for silent communication systems along with machine learning methods (see e.g. <https://doi.org/10.3390/s22010299>).

The claim that “our system captured each word characteristic rather than the individual user’s characteristics” (lines 273-275) is not properly justified. For instance, this could be easily demonstrated by training only on subject A’s data and then testing only on subject B’s data or vice-versa.

The comparison of 3D CNN with SVM is unclear. The reference manuscript [52] used for comparative analysis (lines 383-386) focuses only on changing the final classification layer from a softmax to an SVM. As such, the authors aren’t comparing a conventional SVM to their network as claimed but just comparing different classification layers. Conversely, if they are applying an SVM directly to the raw data then one should definitely expect better results than those reported as the dimension of the data is too high to produce accurate results with an SVM – SVMs are much suited to taking smaller input sizes. In this respect, normally, feature extraction would be done on the data first, rather than applying the model to raw data.

The argument reported in relation to the comparison of the achieved 35% accuracy with state-of-the-art sEMG performance is not accurate. Namely, the claim “Although the state-of-the-art performance with high accuracy (~92%) was demonstrated, the system electrode size was two orders of magnitude larger than that of this work” (lines 464-466) should be referred to in relation to the points of contact. As the authors tailored their testing framework to perform well on their strain gauge setup which takes resistance recordings, it doesn’t seem a fair comparison if a different form of signal data is inputted instead.

The neural network architecture described in the Section “Three-dimensional CNN for SiNM strain gauge signal analysis” doesn’t seem to tally with the dimensions shown in the corresponding figure (Fig 3). The relation between the size of the data at each stage and the kernels being applied should be further clarified.

Finally, the authors should comment on the limitation of the proposed device approach, which is not very user friendly to wear, and justify further the proposed use-case.

Reviewer #2 (Remarks to the Author):

This manuscript presents an approach of identifying verbal communication without vocalization by combining the sensor technology in data measurement and machine learning algorithms in data processing. Strain sensor that was based on SiNM was fabricated and placed on the mouth surroundings to record the strain information via change of electrical resistance when people speak. The measured strain and time correlation were converted to 2D image as machine learning input data. CNN model was used in machine learning model and three different classifier models were used to determine training accuracy. In addition, sEMG sensor was also used to show the merits of SiNM strain sensor measurement. The following issues need to be addressed before I recommend the acceptance for publication.

1. Compared with the direct taking image of mouth during speaking, what’s the key advantage of using strain resistance and time recording (as 2D image input) for machine learning model input? why the image of mouth during speaking was not considered in machine learning model?
2. In comparison of strain sensor that only senses two directions in the current design, sEMG sensor could capture much richer information about speaking process as it is capable of sensing muscle activities from all direction. As signals from sEMG and strain sensor measurement were different, it is questionable that the same machine learning model and training process was use to process them, and it is highly possible that if the accuracy from sEMG with appropriate machine learning model is better than that of strain sensor. justification is needed.
3. In method section, there lacked of detailed data processing steps with machine learning. For example, what’s the criteria for the selection of convolution layers? what the ratio of raw data compared with training and testing data?
4. It seems FEA was done without substrate- which is different with the application environment that was mounted on human skin. It should be improved.

5. A skin-safe pressure-sensitive adhesive was applied to the skin. did the strain sensor was mounted on this adhesive layer or directly on human skin? if it is former, how did this adhesive affect the strain recording activities?

Reviewer #1

General Comment #1: The manuscript is of good quality overall and shows interesting results, with a few shortcomings. It can be published following a revision to clarify the following aspects and add the missing information.

Our response: We thank the reviewer for this positive comment, and the recommendation to publish in *Nature Communications*. We made our revision with pleasure based on the reviewer's opinion, and all the details of our modifications are indicated in our responses and corresponding modifications. All changes are highlighted in the revised version of the manuscript as well.

Comment #1: The comparison between semiconductor and with metal-based strain gauges (lines 238-240) is not sufficiently informative of the state-of-the-art in relation to strain gauges-based systems. It would be useful to compare with other wearable state-of-the-art strain gauges which have already being used for similar purposes, such as graphene-based strain gauges for silent communication systems along with machine learning methods (see e.g. <https://doi.org/10.3390/s22010299>).

Our response: We thank the reviewer for this comment. We compared our ultra-thin crystalline Si based strain gauges with metal-based strain gauges because metal foil is commonly used material for the strain measurement in industry, and we totally agree that the silicon based strain gauges should be compared with other state-of-the-art strain gauges systematically. Hence, we have found papers regarding various silent communication systems utilizing strain gauges including graphene-based system that the reviewer referred above. The detailed comparison with some key parameters such as gauge factor, the number of channels, the number of classes, and recognition accuracy was added in the additional supplementary table. Although these systems recognize various verbal or nonverbal languages, recognition classes are only few simple words/gestures to a dozen of words/gestures. Here, our system demonstrates relatively much higher recognition accuracy (87%) with the incomparably large number of classes (100 verbal words). We strongly believe that this result attributes to the inherent stability and reproducibility of the devices that were fabricated based on inorganic materials with a top-down approach.

Our modification to the manuscript: To support the reviewer's comment, we added Supplementary Table 1 summarizing previously reported silent communication systems using strain gauges, as well as corresponding references. We also added the following sentence to manuscript (line 250-252) "Some of latest silent communication systems based on various strain gauges are compared and summarized in Supplementary Table 1".

Supplementary Table 1. A performance comparison of silent communication systems based on strain gauges

Sensing Technique	Material	Gauge factor	Channel	Attachment position	Language (type)	Class	Accuracy	Reference
Strain	Single crystalline silicon	155	8	Face	English (verbal)	100 words	87.53%	This work
Strain	Graphene	-	1	Throat (Vocal cord)	English + motion (verbal + nonverbal)	15 words + 4 motions	55% (words) 85% (motion)	[1]
Strain	Au nanomesh	7.26 ~ 46.3	18	Face	English vowels (verbal)	3 vowels	-	[2]
Strain	Graphene-coated silk-spandex	19.6 ~ 34.3	10	Fingers	Gesture (nonverbal)	4 gestures	96.07	[3]
Strain + vision	Single-walled carbon nanotube / polydimethylsiloxane	-	5	Hand (Fingers)	Gesture (nonverbal)	10 gestures	100% (normal) 96.7% (noisy)	[4]
Strain	Carbon grease	8	5	Fingers	Sign language (nonverbal)	10 gestures	98%	[5]

Added references

1. Ravenscroft, D. et al. Machine learning methods for automatic silent speech recognition using a wearable graphene strain gauge sensor. *Sensors* **22**, 299 (2021).
2. Wang, Y. et al. A durable nanomesh on-skin strain gauge for natural skin motion monitoring with minimum mechanical constraints. *Science Advances* **6**, eabb7043 (2020).
3. Song, X. et al. A graphene-coated silk-spandex fabric strain sensor for human movement monitoring and recognition. *Nanotechnology* **32**, 215501 (2021).
4. Wang, M. et al. Gesture recognition using a bioinspired learning architecture that integrates visual data with somatosensory data from stretchable sensors. *Nature Electronics* **3**, 563-570 (2020).
5. Li, L., Jiang, S., Shull, P.B. & Gu, G. SkinGest: artificial skin for gesture recognition via filmy stretchable strain sensors. *Advanced Robotics* **32**, 1112-1121 (2018).

Comment #2: The claim that “our system captured each word characteristic rather than the individual user’s characteristics” (lines 273-275) is not properly justified. For instance, this could be easily demonstrated by training only on subject A’s data and then testing only on subject B’s data or vice-versa.

Our response: Thank you for your insightful advice. We apologize for the confusion in understanding the sentence. The intention of the sentence was to state that it is not that there is no user dependency, but there is a correlation between unseen data, so word characteristics can be captured as well by the system. We have modified the sentence into “Our system captured the signal of each user’s word characteristics despite the sensor location dependency and user

dependency” (line 279-281) in the revised manuscript. In addition, as the reviewer suggested, we have trained only on subject B’s data and tested only on A6 datasets, and the accuracy of the unseen user A6 data was 16% (Supplementary Table 9). Interestingly, it was higher than that of training only on A1 datasets which is 6% (Supplementary Table 8). Both results showed low accuracy relatively because they learned only a small amount of unseen data, but it can be interpreted that subject A and subject B have some correlation with each other.

As can be seen in Supplementary Table 8, the accuracy of A6 test is gradually increased as A1~A5 datasets measured on different days were trained. A1~A5 datasets can be regarded as unseen data from the standpoint of A6 datasets since a slight mismatch may occur in the attachment location while reattaching the sensors. Furthermore, Training B which is datasets from unseen users also significantly improves the accuracy of the A6 test. Since each person has a different speaking habit, different accent, and different face shape, the initial accuracy of new users is bound to be low. Nevertheless, these results imply that there are correlations between the unseen data, despite the sensor location dependency and user dependency.

We also conducted additional transfer learning on the model pretrained only with B’s data. It is re-trained with one or two sets among A6. The accuracy increases from 16% to 38%, and then to 68%. This shows that, compared to transfer learning using only training data from subject B and the parts of subject A’s data from unseen sensors, individual customization can be performed effectively as a user continuously accumulates data.

Supplementary Table 8. Result of the unseen data experiment 1

	Training	Test	Accuracy(%)
From Scratch	A1	A6	6
	A1, A2	A6	7
	A1, A2, A3	A6	22
	A1, A2, A3, A4	A6	34
	A1, A2, A3, A4, A5	A6	35
	B + A1, A2, A3, A4, A5 (**)	A6	56
Transfer learning	(**)+ 1 set among A6	A6	83
	(**)+ 2 set among A6	A6	88

Supplementary Table 9. Result of the unseen data experiment 2

	Training	Test	Accuracy(%)
From Scratch	B (*)	A6	16
	B + A1, A2	A6	25
	B + A1, A2	A6	38
	B + A1, A2, A3	A6	47
	B + A1, A2, A3, A4	A6	50
	B + A1, A2, A3, A4, A5 (**)	A6	56
Transfer learning	(*) + 1 set among A6	A6	38
	(*) + 2 set among A6	A6	68
	(**) + 1 set among A6	A6	83
	(**) + 2 set among A6	A6	88

Our modification to the manuscript: We modified the following sentence in the manuscript “Our system captured each word characteristic rather than the individual user’s characteristics” into “Our system captured the signal of each user’s word characteristics despite the sensor location dependency and user dependency” (line 279-281). We added a Supplementary Table 9 which demonstrates the result of the unseen user experiment where training is done on subject B.

Comment #3: The comparison of 3D CNN with SVM is unclear. The reference manuscript [52] used for comparative analysis (lines 383-386) focuses only on changing the final classification layer from a softmax to an SVM. As such, the authors aren’t comparing a conventional SVM to their network as claimed but just comparing different classification layers. Conversely, if they are applying an SVM directly to the raw data then one should definitely expect better results than those reported as the dimension of the data is too high to produce accurate results with an SVM – SVMs are much suited to taking smaller input sizes. In this respect, normally, feature extraction would be done on the data first, rather than applying the model to raw data.

Our response: Thank you for your considerate comment. We referred to the reference #[52] due to the similarities in our works related to a conventional SVM, which is utilized for classification. We added an additional reference to help the readers better understand. We compared our method with SVM in order to demonstrate how we used a deep learning network (e.g., 3D CNN) for the first time to analyze strain gauges. The reference #[12] is cited to

recognize SSRS based on sEMG. It utilizes a machine learning-based network to predict silent speech user intentions. We conducted comparative experiments with existing machine learning-based models based on reference # [12]. As the reviewer mentioned, one can be thought that the raw data dimensions based on conventional SVM are too high to show good performance. We additionally experiment with a deep learning-based SVM network [52] that analyzes the feature map before a fully connected layer. We trained the network with 5-fold cross-validation in the same way as the main experiment, achieving an average accuracy of 86.27%. This result proves that our method, a deep learning-based model, is able to learn representations well because it ultimately optimizes the CNN network. We also trained models using only classical machine learning techniques (LDA, SVM) with input feature size reduction using PCA. The accuracy results of both strain gauge and sEMG with PCA were lower than those without PCA for feature reduction.

Our modification to the manuscript: We added a Supplementary Table 5 which shows comparisons with the conventional methods

Supplementary Table 5. Comparison with the conventional methods

Method	Strain Gauge						sEMG
	Fold 1	Fold 2	Fold 3	Fold 4	Fold 5	Avg	Val set
PCA + LDA	6.4	8.85	16	10.15	11.7	10.62	2.2
PCA + SVM	11.5	15.65	46.85	31.15	22.25	25.48	2.55
SVM	58.8	71.3	76.25	72.7	67.1	69.23	4.9
CONV + SVM	79.95	84.8	90.2	90.55	85.85	86.27	40.85
Transformer	71.85	74.85	66.8	69.4	74.8	71.54	28.25
VGG	67.3	67.1	76.3	74.9	70.2	71.16	22.25
Ours	80.1	87.85	91.55	90.5	87.65	87.53	42.60

Added references

- M. A. Hearst, S. T. Dumais, E. Osuna, J. Platt and B. Scholkopf, "Support vector machines," in *IEEE Intelligent Systems and their Applications*, vol. 13, no. 4, pp. 18-28, July-Aug. 1998, doi: 10.1109/5254.708428.

- Blei, David M., Andrew Y. Ng, and Michael I. Jordan. "Latent dirichlet allocation." *Journal of machine Learning research* 3.Jan (2003): 993-1022.

Comment #4: The argument reported in relation to the comparison of the achieved 35% accuracy with state-of-the-art sEMG performance is not accurate. Namely, the claim “Although the state-of-the-art performance with high accuracy (~92%) was demonstrated, the system electrode size was two orders of magnitude larger than that of this work” (lines 464-466) should be referred to in relation to the points of contact. As the authors tailored their testing framework to perform well on their strain gauge setup which takes resistance recordings, it doesn’t seem a fair comparison if a different form of signal data is inputted instead.

Our response: We thank the reviewer for the constructive comment. The sentence may give some confusion in understanding. We modified the sentence “Although the state-of-the-art performance with high accuracy (~92%) was demonstrated, the system electrode size was two orders of magnitude larger than that of this work” into “The state-of-the-art performance used the system electrode whose size is two orders of magnitude larger than that of this work. To make a fair comparison, we downscaled the sEMG electrodes to be identical to the size of our strain gauges” (line 470-472).

To analyze the performance of two different types of signals, we additionally performed several experiments. We made comparison with PCA, networks in the references #[12, 52], Transformer[*][**], VGGNet[***], and our proposed 3D-CNN model. Unlike using the same architecture for both signals before, we carefully fine-tuned each signal’s network parameters and structures. As a result, we got the best accuracies for both types of signals with our proposed model. It can be interpreted that our method is not just robust to the strain gauge, but also robust to the sEMG. Especially, sEMG performance improved from (from 35% to 42.6%) which is better than the others shown in Supplementary Table 5. However, the result is still far inferior to our strain gauge sensor. The contact area of the device of the skin becomes smaller due to downsizing the physical dimension of sEMG electrode, leading to a low signal-to-noise-ratio of the sEMG signal.

We also evaluated the performance of the two types of signals on the Transformer and VGG networks. The former is commonly used to handle sequential data such as speech/audio, text, and video, and the latter is famous for the classification task. The detailed structures of our 3D-CNN, Transformer, and VGG models are given below. The performances with strain inputs are 87.53%, 71.54% and 71.16% for 3D-CNN, Transformer and VGG, respectively. These results verify the superiority of our strain gauge sensors, and that our original testing framework was not specifically tailored to perform well on our strain gauge sensors.

Our modification to the manuscript: We modified the sentence: “Although the state-of-the-art performance with high accuracy (~92%) was demonstrated, the system electrode size was two orders of magnitude larger than that of this work” into “The state-of-the-art performance used the system electrode whose size is two orders of magnitude larger than that of this work. To make a fair comparison, we downscaled the sEMG electrodes to be identical size of our strain gauges” (Line 470-472). We added the following sentence to the manuscript “The comparison of performance with other models which are commonly used to handle sequential data such as speech/audio, text, and video is provided in Supplementary Table 5” (line 393-395). We also changed the result of sEMG from 35% to 42.6% in the manuscript. We added a Supplementary Table 5 including the results described above, which clarifies the validity of our 3D CNN model. We also added Supplementary Table 10, 11, 12, 13 to provide the detailed

structures, and mentioned it in the Method section with the following sentence “The detailed structure of each model is provided in Supplementary Table 10, 11, 12, 13” (Line 604). Finally, we modified Figure 5g, h according to the changes of performance due to the fine tuning.

Supplementary Table 10. Model details for SiNM-based Strain Gauge

Layer name	Operator	Kernel size	Padding	Stride	Channel size	Ouput Size
Conv 1	$\begin{bmatrix} Conv3d \\ InstanceNorm3d \\ ReLU \\ Dropout3d (0.3) \end{bmatrix}$	3 x 3 x 3	(1, 1, 1)	(1, 1, 2)	32	2 x 4 x 300
Conv 2	$\begin{bmatrix} Conv3d \\ InstanceNorm3d \\ ReLU \\ Dropout3d (0.3) \end{bmatrix}$	3 x 3 x 3	(1, 1, 1)	(1, 1, 2)	32	2 x 4 x 150
	$\begin{bmatrix} Conv3d \\ InstanceNorm3d \\ ReLU \\ Dropout3d (0.3) \end{bmatrix}$	3 x 3 x 3	(1, 1, 1)	(1, 1, 1)		
Conv 3	$\begin{bmatrix} Conv3d \\ InstanceNorm3d \end{bmatrix}$	1 x 3 x 3	(0, 1, 1)	(1, 2, 2)	64	2 x 2 x 75
Conv 4	$\begin{bmatrix} Conv3d \\ InstanceNorm3d \\ ReLU \\ Dropout3d (0.3) \end{bmatrix}$	3 x 3 x 3	(1, 1, 1)	(1, 1, 2)	64	2 x 2 x 38
	$\begin{bmatrix} Conv3d \\ InstanceNorm3d \\ ReLU \\ Dropout3d (0.3) \end{bmatrix}$	3 x 3 x 3	(1, 1, 1)	(1, 1, 1)		
Conv 5	$\begin{bmatrix} Conv3d \\ InstanceNorm3d \end{bmatrix}$	3 x 3 x 3	(1, 1, 1)	(1, 2, 2)	128	2 x 1 x 19
Conv 6	$\begin{bmatrix} Conv3d \\ InstanceNorm3d \\ ReLU \\ Dropout3d (0.3) \end{bmatrix}$	3 x 3 x 3	(1, 1, 1)	(1, 1, 2)	128	2 x 1 x 10
	$\begin{bmatrix} Conv3d \\ InstanceNorm3d \\ ReLU \\ Dropout3d (0.3) \end{bmatrix}$	3 x 3 x 3	(1, 1, 1)	(1, 1, 1)		
Conv 7	$\begin{bmatrix} Conv3d \\ InstanceNorm3d \end{bmatrix}$	3 x 3 x 3	(1, 1, 1)	(1, 2, 2)	256	2 x 1 x 5
Conv 8	$\begin{bmatrix} Conv3d \\ InstanceNorm3d \\ ReLU \\ Dropout3d (0.3) \end{bmatrix}$	3 x 3 x 3	(1, 1, 1)	(1, 1, 2)	256	2 x 1 x 3

	$\begin{bmatrix} Conv3d \\ InstanceNorm3d \\ ReLU \\ Dropout3d (0.3) \end{bmatrix}$	3 x 3 x 3	(1, 1, 1)	(1, 1, 1)		
FC 1	Linear	-				512
FC 2	Linear	-				100
FC 3	Linear	-				100

Supplementary Table 11. Model details for sEMG

Layer name	Operator	Kernel size	Padding	Stride	Channel size	Output Size
Conv 1	$\begin{bmatrix} Conv3d \\ InstanceNorm3d \\ ReLU \\ Dropout3d (0.3) \end{bmatrix}$	3 x 3 x 3	(1, 1, 1)	(1, 1, 2)	32	1 x 4 x 1000
Conv 2	$\begin{bmatrix} Conv3d \\ InstanceNorm3d \\ ReLU \\ Dropout3d (0.3) \end{bmatrix}$	3 x 3 x 3	(1, 1, 1)	(1, 1, 2)	32	1 x 4 x 500
	$\begin{bmatrix} Conv3d \\ InstanceNorm3d \\ ReLU \\ Dropout3d (0.3) \end{bmatrix}$	3 x 3 x 3	(1, 1, 1)	(1, 1, 1)		
Conv 3	$\begin{bmatrix} Conv3d \\ InstanceNorm3d \end{bmatrix}$	1 x 3 x 3	(0, 1, 1)	(1, 2, 2)	64	1 x 2 x 250
Conv 4	$\begin{bmatrix} Conv3d \\ InstanceNorm3d \\ ReLU \\ Dropout3d (0.3) \end{bmatrix}$	3 x 3 x 3	(1, 1, 1)	(1, 1, 2)	64	1 x 2 x 125
	$\begin{bmatrix} Conv3d \\ InstanceNorm3d \\ ReLU \\ Dropout3d (0.3) \end{bmatrix}$	3 x 3 x 3	(1, 1, 1)	(1, 1, 1)		
Conv 5	$\begin{bmatrix} Conv3d \\ InstanceNorm3d \end{bmatrix}$	3 x 3 x 3	(1, 1, 1)	(1, 2, 2)	128	1 x 1 x 63
Conv 6	$\begin{bmatrix} Conv3d \\ InstanceNorm3d \\ ReLU \\ Dropout3d (0.3) \end{bmatrix}$	3 x 3 x 3	(1, 1, 1)	(1, 1, 2)	128	1 x 1 x 32

	$\begin{bmatrix} Conv3d \\ InstanceNorm3d \\ ReLU \\ Dropout3d (0.3) \end{bmatrix}$	3 x 3 x 3	(1, 1, 1)	(1, 1, 1)		
Conv 7	$\begin{bmatrix} Conv3d \\ InstanceNorm3d \end{bmatrix}$	3 x 3 x 3	(1, 1, 1)	(1, 2, 2)	256	1 x 1 x 16
Conv 8	$\begin{bmatrix} Conv3d \\ InstanceNorm3d \\ ReLU \\ Dropout3d (0.3) \end{bmatrix}$	3 x 3 x 3	(1, 1, 1)	(1, 1, 2)	256	1 x 1 x 8
	$\begin{bmatrix} Conv3d \\ InstanceNorm3d \\ ReLU \\ Dropout3d (0.3) \end{bmatrix}$	3 x 3 x 3	(1, 1, 1)	(1, 1, 1)		
FC 1	Linear	-				512
FC 2	Linear	-				100
FC 3	Linear	-				100

Supplementary Table 12. Detailed structure of Transformer

Layer name	Operator	Kernel Size	Padding	Stride	Channel Size	Output Size
Conv 1	$\begin{bmatrix} Conv1d \\ ReLU \\ Batchnorm \end{bmatrix}$	5	2	2	32	300
Conv 2	$\begin{bmatrix} Conv1d \\ ReLU \\ Dropout (0.2) \end{bmatrix}$	5	2	2	64	150
Conv 3	$\begin{bmatrix} Conv1d \\ ReLU \\ Batchnorm \end{bmatrix}$	5	2	2	128	75
Conv 4	$\begin{bmatrix} Conv1d \\ ReLU \\ Dropout (0.2) \end{bmatrix}$	5	2	2	256	38
Transformer encoder 1	-				256	38
Transformer encoder 2	-				256	38
Self-attention pooling						256
FC 1	$\begin{bmatrix} Linear \\ ReLU \end{bmatrix}$	-				400
FC 2	$\begin{bmatrix} Linear \\ ReLU \end{bmatrix}$	-				400

FC 3	$\begin{bmatrix} \text{Linear} \\ \text{ReLU} \end{bmatrix}$	-	400
FC 4	Linear	-	100

Supplementary Table 13. Detailed structure of VGG

Layer name	Operator	Kernel Size	Padding	Stride	Channel Size	Output Size
Conv 1	Conv2d	(3, 7)	(1, 3)	(1, 2)	64	8 x 300
Conv 2	$\begin{bmatrix} \text{Conv2d} \\ \text{Batchnorm} \\ \text{ReLU} \\ \text{Maxpool} \end{bmatrix}$	(3, 3)	(1, 1)	(1, 1)	64	4 x 150
Conv 3	$\begin{bmatrix} \text{Conv2d} \\ \text{Batchnorm} \\ \text{ReLU} \\ \text{Maxpool} \end{bmatrix}$	(3, 3)	(1, 1)	(1, 1)	128	2 x 75
Conv 4	$\begin{bmatrix} \text{Conv2d} \\ \text{Batchnorm} \\ \text{ReLU} \end{bmatrix}$	(3, 3)	(1, 1)	(1, 1)	256	1 x 37
	$\begin{bmatrix} \text{Conv2d} \\ \text{Batchnorm} \\ \text{ReLU} \\ \text{Maxpool} \end{bmatrix}$	(3, 3)	(1, 1)	(1, 1)	256	
Conv 5	$\begin{bmatrix} \text{Conv2d} \\ \text{Batchnorm} \\ \text{ReLU} \end{bmatrix}$	(1, 3)	(0, 1)	(1, 1)	512	1 x 18
	$\begin{bmatrix} \text{Conv2d} \\ \text{Batchnorm} \\ \text{ReLU} \\ \text{Maxpool} \end{bmatrix}$	(1, 3)	(0, 1)	(1, 1)	512	
Conv 6	$\begin{bmatrix} \text{Conv2d} \\ \text{Batchnorm} \\ \text{ReLU} \end{bmatrix}$	(1, 3)	(0, 1)	(1, 1)	512	1 x 9
	$\begin{bmatrix} \text{Conv2d} \\ \text{Batchnorm} \\ \text{ReLU} \\ \text{Maxpool} \end{bmatrix}$	(1, 3)	(0, 1)	(1, 1)	512	
Statistic pooling						1024
FC 1	$\begin{bmatrix} \text{Linear} \\ \text{ReLU} \\ \text{Dropout}(0.65) \end{bmatrix}$				-	4096
FC 2	$\begin{bmatrix} \text{Linear} \\ \text{ReLU} \\ \text{Dropout}(0.65) \end{bmatrix}$				-	4096
FC 3	Linear				-	100

Modified Figure 5g, h

Added references

- [*] Safari, Pooyan, Miquel India, and Javier Hernando. "Self-attention encoding and pooling for speaker recognition." *arXiv preprint arXiv:2008.01077* (2020).
- [**] Vaswani, Ashish, et al. "Attention is all you need." *Advances in neural information processing systems* 30 (2017).
- [***] Simonyan, Karen, and Andrew Zisserman. "Very deep convolutional networks for large-scale image recognition." *arXiv preprint arXiv:1409.1556* (2014).

Comment #5: The neural network architecture described in the Section “Three-dimensional CNN for SiNM strain gauge signal analysis” doesn’t seem to tally with the dimensions shown in the corresponding figure (Fig 3). The relation between the size of the data at each stage and the kernels being applied should be further clarified.

Our response: We thank the reviewer for this comment. We agree with your comment that our figure might be unclear. We believe the readers may have confused the information about the size of the input data and the channel information of the feature map. The size of the input tensor to our model is 2x4x600 with one channel, and 3D feature maps with several channels are generated through 3D convolution layers. The details of the architecture are shown in the modified Fig. 3a and Supplementary Table 10, 11.

Our modification to the manuscript: We modified our main Figure 3a to describe our deep learning architecture more precisely. We also added Supplementary Table 10, 11 demonstrating the details of the architecture such as kernel size, number of channels, input/output size, etc.

Modified Figure 3a

Comment #6: Finally, the authors should comment on the limitation of the proposed device approach, which is not very user friendly to wear, and justify further the proposed use-case.

Our response: We thank the reviewer for this comment. We agree that the device wearing approach is not somewhat user friendly, and we have not showed the detailed device wearing procedure. However, the main purpose of this work focuses mainly on implementing a novel silent speech interface that is capable of classifying a large number of classes (100 words) utilizing scaled-down strain gauges for the first time. Therefore, we have not really focused on the device wearability. However, as the reviewer mentioned, it is worth to show how our device can be mounted on the skin. From the perspective of its wearability, we explained the whole sensor attachment process in Method (line 567-572) with some additional sentences. Furthermore, we added a supplementary video, visualizing the sensor attachment process to make the readers better understand the device wearability. Based on our experiment for the device wearability, our device can extremely conformally contact to the facial skin. By observing this, we can realize that the device is capable of precisely detecting the facial movement dynamics.

Our modification to the manuscript: We modified the sentences in Methods section “A skin-safe pressure-sensitive adhesive (Derma-tac from Smooth-On) was applied to the designated position, which was selected through the preliminary study presented in Supplementary Figs. 1 and 2. Water-soluble tapes with our strain sensor transferred on were attached to the position with moderate pressure, and DI water was then gently sprayed using a dispenser for 1 min to dissolve the PVA film.” into the following sentences “A skin-safe pressure-sensitive adhesive (Derma-tac from Smooth-On) was applied to the backside of the sensors mounted on water-soluble tapes. And then, the strains were attached to the position which was selected through the preliminary study presented in Supplementary Figs. 1 and 2 with moderate pressure, and DI water was then gently sprayed using a dispenser for 1 min to dissolve the PVA film” (Line 568-572). We also added a Supplementary Video 3 ‘SensorAttachment’ featuring the device attachment process.

Supplementary Video 3: please check the attached video file ‘SensorAttachment’ including whole process of wearing our device to the facial skin.

Reviewer #2

General comment #1: This manuscript presents an approach of identifying verbal communication without vocalization by combining the sensor technology in data measurement and machine learning algorithms in data processing. Strain sensor that was based on SiNM was fabricated and placed on the mouth surroundings to record the strain information via change of electrical resistance when people speak. The measured strain and time correlation were converted to 2D image as machine learning input data. CNN model was used in machine learning model and three different classifier models were used to determine training accuracy. In addition, sEMG sensor was also used to show the merits of SiNM strain sensor measurement. The following issues need to be addressed before I recommend the acceptance for publication.

Our response: We thank reviewer for summing up our manuscript and presenting some issues to be addressed. We agree with the reviewer's request for change and reflected the criticisms as much as possible for the better readership of readers. The corresponding revisions to each comment are indicated in response. All the details of our modifications are indicated in our responses and corresponding modifications. All changes are highlighted in the revised version of the manuscript as well.

Comment #1: Compared with the direct taking image of mouth during speaking, what's the key advantage of using strain resistance and time recording (as 2D image input) for machine learning model input? why the image of mouth during speaking was not considered in machine learning model?

Our response: We thank the reviewer for this opinion first. We think that vision input and strain input both have their pros and cons. As the reviewer mentioned, the vision input (image) has incomparably high spatial resolution than the other inputs from wearable sensors such as sEMG or strain gauge. However, we already highlighted the utility as a key advantage of using wearable platforms in the Introduction section. For use in static situations, vision recognition with a camera may provide high word recognition accuracy only if the subject is directly looking straight at the camera under the limited condition where light is sufficiently provided. For this reason, however, it is not always usable in daily life when dynamic motions of the subjects are required, and the subjects do not possess the cameras. On the other hand, in the case of wearable platforms, the spatial resolution is lower than the vision registration, but it can be used in more diverse situations because the subjects are not required to possess the camera or staring at the camera with a relatively bright environment. From this point of view, we thought that strain input has novelty in that it has potential to significantly improve spatial resolution over sEMG input. We realized that our explanation of these arguments was insufficient, and we modified some sentences for better readership of the readers.

Our modification to the manuscript: We replaced the sentence “Nevertheless, the continuous shooting of the face in a static environment is indispensable to avoid an accuracy drop by body motion or light-induced artifacts, which leads to user inconvenience for daily communication routines.” with the following sentences “However, there are many situations in which the daily use of vision recognition is limited since the continuous shooting of the face in a static environment is indispensable. Changes in the shooting direction due to body motion and

changes in the light intensity according to the surrounding environment can lead to a significant drop in recognition accuracy. Furthermore, unnecessary information such as background may occupy more pixels than speech-related information, which can be an obstruction in analyzing speech information” (Line 206-213).

Comment #2: In comparison of strain sensor that only senses two directions in the current design, sEMG sensor could capture much richer information about speaking process as it is capable of sensing muscle activities from all direction. As signals from sEMG and strain sensor measurement were different, it is questionable that the same machine learning model and training process was used to process them, and it is highly possible that if the accuracy from sEMG with appropriate machine learning model is better than that of strain sensor. justification is needed.

Our response: Thank you for the reviewer’s considerate comment. Our main intention in comparing sEMG and strain input was to feature the potential of the strain gauges toward high-resolution systems. Unlike using the same architecture for both signals before, we carefully fine-tuned the network parameters and structures for each signal, which increases the accuracy from 35% to 42.6%. We used the same deep learning model with different details, and the details were added in Supplementary Table 10, 11.

For looking forward to finding a more appropriate machine learning model, we additionally performed several experiments. We made comparison with PCA, networks in the reference #[12, 52], Transformer, VGGNet and our proposed 3D-CNN model. As a result, we got the best accuracies for both types of signals with our proposed model, as shown in Supplementary Table 5. It shows that our model not only performed effectively on strain gauge but also performed well on sEMG. However, sEMG result is still inferior to our strain gauge system mainly due to the downscaling of sEMG electrode as much as our strain gauge, leading to a low signal-to-noise ratio of the EMG signal.

Our modification to the manuscript: We added Supplementary Table 5 for an extra experiment to compare with the other networks. Furthermore, We added Supplementary Table 10, 11 to clarify the model details we used for our system and sEMG system. We also added Supplementary Table 12, 13 for the detailed information of Transformer and VGGNet, which are used for comparison. We added the following sentence about these tables to the Method section “The detailed structure of each model is provided in Supplementary Table 10, 11, 12, 13” (Line 604). Finally, we modified Figure 5g, h according to the performance changes due to the fine tuning.

Supplementary Table 5. Comparison with the conventional methods

Method	Strain Gauge						sEMG
	Fold 1	Fold 2	Fold 3	Fold 4	Fold 5	Avg	Val set
PCA + LDA	6.4	8.85	16	10.15	11.7	10.62	2.2

PCA + SVM	11.5	15.65	46.85	31.15	22.25	25.48	2.55
SVM	58.8	71.3	76.25	72.7	67.1	69.23	4.9
CONV + SVM	79.95	84.8	90.2	90.55	85.85	86.27	40.85
Transformer	71.85	74.85	66.8	69.4	74.8	71.54	28.25
VGG	67.3	67.1	76.3	74.9	70.2	71.16	22.25
Ours	80.1	87.85	91.55	90.5	87.65	87.53	42.60

Supplementary Table 10. Model details for SiNM-based Strain Gauge

Layer name	Operator	Kernel size	Padding	Stride	Channel size	Ouput Size
Conv 1	$\begin{bmatrix} Conv3d \\ InstanceNorm3d \\ ReLU \\ Dropout3d (0.3) \end{bmatrix}$	3 x 3 x 3	(1, 1, 1)	(1, 1, 2)	32	2 x 4 x 300
Conv 2	$\begin{bmatrix} Conv3d \\ InstanceNorm3d \\ ReLU \\ Dropout3d (0.3) \end{bmatrix}$	3 x 3 x 3	(1, 1, 1)	(1, 1, 2)	32	2 x 4 x 150
	$\begin{bmatrix} Conv3d \\ InstanceNorm3d \\ ReLU \\ Dropout3d (0.3) \end{bmatrix}$	3 x 3 x 3	(1, 1, 1)	(1, 1, 1)		
Conv 3	$\begin{bmatrix} Conv3d \\ InstanceNorm3d \end{bmatrix}$	1 x 3 x 3	(0, 1, 1)	(1, 2, 2)	64	2 x 2 x 75
Conv 4	$\begin{bmatrix} Conv3d \\ InstanceNorm3d \\ ReLU \\ Dropout3d (0.3) \end{bmatrix}$	3 x 3 x 3	(1, 1, 1)	(1, 1, 2)	64	2 x 2 x 38
	$\begin{bmatrix} Conv3d \\ InstanceNorm3d \\ ReLU \\ Dropout3d (0.3) \end{bmatrix}$	3 x 3 x 3	(1, 1, 1)	(1, 1, 1)		
Conv 5	$\begin{bmatrix} Conv3d \\ InstanceNorm3d \end{bmatrix}$	3 x 3 x 3	(1, 1, 1)	(1, 2, 2)	128	2 x 1 x 19
Conv 6	$\begin{bmatrix} Conv3d \\ InstanceNorm3d \\ ReLU \\ Dropout3d (0.3) \end{bmatrix}$	3 x 3 x 3	(1, 1, 1)	(1, 1, 2)	128	2 x 1 x 10

	$\begin{bmatrix} Conv3d \\ InstanceNorm3d \\ ReLU \\ Dropout3d (0.3) \end{bmatrix}$	3 x 3 x 3	(1, 1, 1)	(1, 1, 1)		
Conv 7	$\begin{bmatrix} Conv3d \\ InstanceNorm3d \end{bmatrix}$	3 x 3 x 3	(1, 1, 1)	(1, 2, 2)	256	2 x 1 x 5
Conv 8	$\begin{bmatrix} Conv3d \\ InstanceNorm3d \\ ReLU \\ Dropout3d (0.3) \end{bmatrix}$	3 x 3 x 3	(1, 1, 1)	(1, 1, 2)	256	2 x 1 x 3
	$\begin{bmatrix} Conv3d \\ InstanceNorm3d \\ ReLU \\ Dropout3d (0.3) \end{bmatrix}$	3 x 3 x 3	(1, 1, 1)	(1, 1, 1)		
FC 1	Linear		-			512
FC 2	Linear		-			100
FC 3	Linear		-			100

Supplementary Table 11. Model details for sEMG

Layer name	Operator	Kernel size	Padding	Stride	Channel size	Output Size
Conv 1	$\begin{bmatrix} Conv3d \\ InstanceNorm3d \\ ReLU \\ Dropout3d (0.3) \end{bmatrix}$	3 x 3 x 3	(1, 1, 1)	(1, 1, 2)	32	1 x 4 x 1000
Conv 2	$\begin{bmatrix} Conv3d \\ InstanceNorm3d \\ ReLU \\ Dropout3d (0.3) \end{bmatrix}$	3 x 3 x 3	(1, 1, 1)	(1, 1, 2)	32	1 x 4 x 500
	$\begin{bmatrix} Conv3d \\ InstanceNorm3d \\ ReLU \\ Dropout3d (0.3) \end{bmatrix}$	3 x 3 x 3	(1, 1, 1)	(1, 1, 1)		
Conv 3	$\begin{bmatrix} Conv3d \\ InstanceNorm3d \end{bmatrix}$	1 x 3 x 3	(0, 1, 1)	(1, 2, 2)	64	1 x 2 x 250
Conv 4	$\begin{bmatrix} Conv3d \\ InstanceNorm3d \\ ReLU \\ Dropout3d (0.3) \end{bmatrix}$	3 x 3 x 3	(1, 1, 1)	(1, 1, 2)	64	1 x 2 x 125

	$\begin{bmatrix} \text{Conv3d} \\ \text{InstanceNorm3d} \\ \text{ReLU} \\ \text{Dropout3d (0.3)} \end{bmatrix}$	3 x 3 x 3	(1, 1, 1)	(1, 1, 1)		
Conv 5	$\begin{bmatrix} \text{Conv3d} \\ \text{InstanceNorm3d} \end{bmatrix}$	3 x 3 x 3	(1, 1, 1)	(1, 2, 2)	128	1 x 1 x 63
Conv 6	$\begin{bmatrix} \text{Conv3d} \\ \text{InstanceNorm3d} \\ \text{ReLU} \\ \text{Dropout3d (0.3)} \end{bmatrix}$	3 x 3 x 3	(1, 1, 1)	(1, 1, 2)	128	1 x 1 x 32
	$\begin{bmatrix} \text{Conv3d} \\ \text{InstanceNorm3d} \\ \text{ReLU} \\ \text{Dropout3d (0.3)} \end{bmatrix}$	3 x 3 x 3	(1, 1, 1)	(1, 1, 1)		
Conv 7	$\begin{bmatrix} \text{Conv3d} \\ \text{InstanceNorm3d} \end{bmatrix}$	3 x 3 x 3	(1, 1, 1)	(1, 2, 2)	256	1 x 1 x 16
Conv 8	$\begin{bmatrix} \text{Conv3d} \\ \text{InstanceNorm3d} \\ \text{ReLU} \\ \text{Dropout3d (0.3)} \end{bmatrix}$	3 x 3 x 3	(1, 1, 1)	(1, 1, 2)	256	1 x 1 x 8
	$\begin{bmatrix} \text{Conv3d} \\ \text{InstanceNorm3d} \\ \text{ReLU} \\ \text{Dropout3d (0.3)} \end{bmatrix}$	3 x 3 x 3	(1, 1, 1)	(1, 1, 1)		
FC 1	Linear	-				512
FC 2	Linear	-				100
FC 3	Linear	-				100

Supplementary Table 12. Detailed structure of Transformer

Layer name	Operator	Kernel Size	Padding	Stride	Channel Size	Output Size
Conv 1	$\begin{bmatrix} \text{Conv1d} \\ \text{ReLU} \\ \text{Batchnorm} \end{bmatrix}$	5	2	2	32	300
Conv 2	$\begin{bmatrix} \text{Conv1d} \\ \text{ReLU} \\ \text{Dropout (0.2)} \end{bmatrix}$	5	2	2	64	150
Conv 3	$\begin{bmatrix} \text{Conv1d} \\ \text{ReLU} \\ \text{Batchnorm} \end{bmatrix}$	5	2	2	128	75
Conv 4	$\begin{bmatrix} \text{Conv1d} \\ \text{ReLU} \\ \text{Dropout (0.2)} \end{bmatrix}$	5	2	2	256	38

Transformer encoder 1		-	256	38
Transformer encoder 2		-	256	38
Self-attention pooling				256
FC 1	[Linear] [ReLU]		-	400
FC 2	[Linear] [ReLU]		-	400
FC 3	[Linear] [ReLU]		-	400
FC 4	Linear		-	100

Supplementary Table 13. Detailed structure of VGG

Layer name	Operator	Kernel Size	Padding	Stride	Channel Size	Output Size
Conv 1	Conv2d	(3, 7)	(1, 3)	(1, 2)	64	8 x 300
Conv 2	[Conv2d] [Batchnorm] [ReLU] [Maxpool]	(3, 3)	(1, 1)	(1, 1)	64	4 x 150
Conv 3	[Conv2d] [Batchnorm] [ReLU] [Maxpool]	(3, 3)	(1, 1)	(1, 1)	128	2 x 75
Conv 4	[Conv2d] [Batchnorm] [ReLU]	(3, 3)	(1, 1)	(1, 1)	256	1 x 37
	[Conv2d] [Batchnorm] [ReLU] [Maxpool]	(3, 3)	(1, 1)	(1, 1)	256	
Conv 5	[Conv2d] [Batchnorm] [ReLU]	(1, 3)	(0, 1)	(1, 1)	512	1 x 18
	[Conv2d] [Batchnorm] [ReLU] [Maxpool]	(1, 3)	(0, 1)	(1, 1)	512	
Conv 6	[Conv2d] [Batchnorm] [ReLU]	(1, 3)	(0, 1)	(1, 1)	512	1 x 9
	[Conv2d] [Batchnorm] [ReLU] [Maxpool]	(1, 3)	(0, 1)	(1, 1)	512	

Statistic pooling			1024
FC 1	$\begin{bmatrix} \text{Linear} \\ \text{ReLU} \\ \text{Dropout}(0.65) \end{bmatrix}$	-	4096
FC 2	$\begin{bmatrix} \text{Linear} \\ \text{ReLU} \\ \text{Dropout}(0.65) \end{bmatrix}$	-	4096
FC 3	Linear	-	100

Modified Figure 5g, h

Comment #3: In method section, there lacked of detailed data processing steps with machine learning. For example, what’s the criteria for the selection of convolution layers? what the ratio of raw data compared with training and testing data?

Our response: We agree with the reviewer that the details of the data processing steps were not sufficiently described in our manuscript. First, in response to the reviewer’s question about “What’s the criteria for the selection of convolution layers?”, we designed our model similarly to VGGNet [*]; a traditional CNN structure. We adopted 3D convolution to learn spatial relationships between adjacent sensors, and excluded the max-pooling layer because of its small input size. The details of the architecture are described in the modified Fig. 3a and Supplementary Table 10, 11. Second, to address the reviewer’s question about “What is the ratio of raw data compared with training and testing data”, we used a train-test split of 80-20, as shown in Fig. 3b.

Our modification to the manuscript: We modified our main Fig. 3a. which describes our deep learning architecture more precisely. We also added Supplementary Tables 10, 11 demonstrating the details of the architecture.

Added references

- [*] Simonyan, Karen, and Andrew Zisserman. "Very deep convolutional networks for large-scale image recognition." *arXiv preprint arXiv:1409.1556* (2014).

Comment #4: It seems FEA was done without substrate- which is different with the application environment that was mounted on human skin. It should be improved.

Our response: We agree with the reviewer's comment that FEA simulation should be conducted in the same state as the actual application environment. According to the previous studies, human skin has comparably low young's modulus of few hundreds kPa[x], [y]. Therefore, we had used PDMS layer, which mimics the human skin with young's modulus of ~100 kPa according to the FEA simulation. We had included the information about PDMS in Supplementary Table 2, but we had mistakenly excluded mentioning it in Supplementary Note 1. We highly appreciate your comment, and we clarified the sentence on that point. We listed the specific parameters of substrate used in FEA simulation to mimic the skin in Supplementary Note 1 and Supplementary Table 2. For better readership, we specified the layers of FEA model used in this work.

Our modification to the manuscript: We modified the sentence in Supplementary Note 1. Details in FEA simulation for better readership "Without loss of generality, the multilayered structures on the PDMS substrate (i.e., PI/Au/Cr/PI/PDMS and PI/Si/PI/PDMS) were modeled using a composite 2D shell with an effective Young's modulus of $E_{effective} = \sum E_i h_i / \sum h_i$, where E_i and h_i are the Young's modulus and thickness of each layer, respectively (Supplementary Table 2).".

Comment #5: A skin-safe pressure-sensitive adhesive was applied to the skin. did the strain sensor was mounted on this adhesive layer or directly on human skin? if it is former, how did this adhesive affect the strain recording activities?

Our response: We highly appreciate your point. We found that the sentence we wrote previously in Method Section is somehow misleading. During the device mounting process onto the skin, we applied the adhesive (Derma-tac from Smooth-On) to the backside of our device with a makeup brush before the device was attached to the skin. We used Derma-tac adhesive to form a better adhesion between the device and the skin and long-term usage. However, it doesn't affect the strain recording since thin adhesive layer does not disturb recording conformal facial movements. On the other hand, however, in the sEMG case, the adhesive layer, which is basically an insulator, closes the conducting path that leads to the distortion of obtained signal. Our strain gauges were double-sided encapsulated without any need for opening the sensing pad like sEMG to make direct contact to the skin. Besides, the sensing pads in sEMG degrades over time due to sebum and sweat. Therefore, the long-term wearability of sEMG is quite limited. Furthermore, the viscosity of Derma-tac adhesive is very low, resulting in a negligible thickness after it is totally dried. We added a supplementary video showing the whole process of attaching our device to skin for readers' better understanding. Additionally, we added a figure which demonstrates a thickness of Derma-tac, and a short video which shows comparison between the sensors attached on PDMS substrate w/ and w/o Derma-tac during stretching for the reviewer's information.

Our modification to the manuscript: We replaced the sentences in the manuscript “A skin-safe pressure-sensitive adhesive (Derma-tac from Smooth-On) was applied to the designated position, which was selected through the preliminary study presented in Supplementary Figs. 1 and 2. Water-soluble tapes with our strain sensor transferred on were attached to the position with moderate pressure, and DI water was then gently sprayed using a dispenser for 1 min to dissolve the PVA film.” with the following sentences “A skin-safe pressure-sensitive adhesive (Derma-tac from Smooth-On) was applied to the backside of the sensors on water-soluble tapes. And then, the sensors were attached to the position which was selected through the preliminary study presented in Supplementary Figs. 1 and 2 with moderate pressure, and DI water was then gently sprayed using a dispenser for 1 min to dissolve the PVA film” (Line 568-572). We also added a Supplementary Video 3 which shows whole process to wear our strain gauges, and correspondingly, we added the following sentence to the manuscript “The whole device attachment process is demonstrated in Supplementary Video 3” (Line 573-574).

Supplementary Video 3: please check the attached video file ‘SensorAttachment’ including whole process of wearing our device to the facial skin.

The thickness of Derma-tac (alpha-step)

Adhesion comparison between sensors attached w/ (right) and w/o (left) Derma-tac: Please check the attached video file ‘Effect of Derma-tac’. As shown in this video, the sensor without Derma-tac was detached from the substrate especially at the bottom non-mesh area where the contact pads are located.

REVIEWER COMMENTS

Reviewer #2 (Remarks to the Author):

The authors addressed most my questions except # 5. Firstly, It is still very confusing whether adhesive was used between sEMG and skin, and please clearly state that. Secondly, Assume that sEMG or strain sensors will be used for a long-time wear (several hours at least) in silent communication systems, and how the degradation of adhesive will affect their performance needs to be evaluated. Thirdly, it is unclear what the plot in "Adhesion comparison between sensors attached w/ (right) and w/o (left) Derma-tac" present with both x and y- axes in length unit. further explanations in details are needed.

Comment #1: Firstly, it is still very confusing whether adhesive was used between sEMG and skin, and please clearly state that.

Our response: We thank the reviewer for this comment and apologize for any confusion this may have caused. To point out the reviewer's comment, we didn't use an additional adhesive layer between sEMG and skin when attaching the sEMG to the face because it blocks the conducting path of the sEMG signal. Instead, the sEMG sensor was attached to the skin using a transparent dressing film (3M Tegaderm) that serves as an adhesive layer and applies to the back side of the sEMG sensor. To make the electrode side open and in direct contact with the skin, a 2-step transfer method using water-soluble tape as a temporary carrier was used. The detailed attachment process and figures of attached sensors are exhibited in the Method section (Experimental process of sEMG DAQ) and Supplementary Fig.7, respectively. This method of attaching sEMG to the face using Tegaderm is referenced by the sEMG article which featured another speech recognition system¹.

Our modification to the manuscript: For better readership, we added the following sentence to clearly state that there are no other materials on the exposed sEMG electrode after transferred to Tegaderm (line 586-588) "By using Tegaderm, a transparent dressing adhesive film directly attached onto the back of the sEMG sensor, the sEMG electrode areas located on the front side are fully opened to make direct contact to the skin while elsewhere is firmly attached to the skin."

Comment #2: Secondly, assume that sEMG or strain sensors will be used for a long-time wear (several hours at least) in silent communication systems, and how the degradation of adhesive will affect their performance needs to be evaluated.

Our response: We appreciate this valuable comment. First, in the data acquisition process, we continuously measured the data about 1000-1500 words with one attachment, which took about 3 hours. It had no effect on the adhesion of Derma-tac and the device performance. To further support this result, we conducted an additional cyclic stretching test using an automatic bending machine with the sensor attached to an elastomer substrate (PDMS) using Derma-tac. The results of about 10,000 cyclic bending tests for 10 hours are shown in the graph below.

As shown in this graph, even after 10 hours, the sensor showed a negligible change in initial resistance within 0.1% except for some temporary fluctuations caused by temperature changes in the experimental environment. This ensures that the device shows reasonable consistency and longevity while performing the stretching test and the silent speech test.

Comment #3: Thirdly, it is unclear what the plot in "Adhesion comparison between sensors attached w/ (right) and w/o (left) Derma-tac" present with both x and y- axes in length unit.

Our response: We thank the reviewer about this comment. We believe that the graph below, which we attached in the previous revision letter, may have been misleading.

The thickness of Derma-tac (Alpha-Step)

We apologize for missing a clear description of this plot, but this is not for "Adhesion comparison between sensors attached w/ (right) and w/o (left) Derma-tac" paragraph. This graph shows the thickness measurements of Derma-tac using a surface profiler, Alpha-Step. We measured the thickness of Derma-tac by moving the probe from the non-Derma-tac area to the Derma-tac area on the wafer substrate. Our intention was to demonstrate that Derma-tac is very thin, $\sim 2\mu\text{m}$, to minimize issues caused by the thickness of the adhesive. We apologize again for any confusion as we did not specify it exactly.

1. Wang, Y. et al. npj Flexible Electronics 5, 1-9 (2021).

REVIEWERS' COMMENTS

Reviewer #2 (Remarks to the Author):

The authors have addressed my questions and improved the manuscript. I recommend the acceptance in the current form.

Comment: The authors have addressed my questions and improved the manuscript. I recommend the acceptance in the current form.

Our response: We thank the reviewer for this comment. We are very pleased that our previous revisions have addressed all the reviewer's questions in an appropriate way. We would gladly share all the data and code generated in this study for a readers' broader understanding. In this revision, we have slightly modified the format of all files according to the author guidance and added some missing statements including Author Contribution, Human Participants Ethics, and Data Availability to make this study suitable for publication. We thank the reviewer again for time and consideration.